# LANTERN: ACCELERATING VISUAL AUTOREGRESSIVE MODELS WITH RELAXED SPECULATIVE DECODING

**Doohyuk Jang**[1*]    **Sihwan Park**[1*]    **June Yong Yang**[1]    **Yeonsung Jung**[1]
**Jihun Yun**[1]    **Souvik Kundu**[2]    **Sungyub Kim**[1†]    **Eunho Yang**[1,3†]
[1]KAIST    [2]Intel Labs    [3]AITRICS
{jadohu, sihwan.park, sungyub.kim, eunhoy}@kaist.ac.kr

## ABSTRACT

Auto-Regressive (AR) models have recently gained prominence in image generation, often matching or even surpassing the performance of diffusion models. However, one major limitation of AR models is their sequential nature, which processes tokens one at a time, slowing down generation compared to models like GANs or diffusion-based methods that operate more efficiently. While speculative decoding has proven effective for accelerating LLMs by generating multiple tokens in a single forward, its application in visual AR models remains largely unexplored. In this work, we identify a challenge in this setting, which we term *token selection ambiguity*, wherein visual AR models frequently assign uniformly low probabilities to tokens, hampering the performance of speculative decoding. To overcome this challenge, we propose a relaxed acceptance condition referred to as LANTERN that leverages the interchangeability of tokens in latent space. This relaxation restores the effectiveness of speculative decoding in visual AR models by enabling more flexible use of candidate tokens that would otherwise be prematurely rejected. Furthermore, by incorporating a total variation distance bound, we ensure that these speed gains are achieved without significantly compromising image quality or semantic coherence. Experimental results demonstrate the efficacy of our method in providing a substantial speed-up over speculative decoding. In specific, compared to a naïve application of the state-of-the-art speculative decoding, LANTERN increases speed-ups by $1.75\times$ and $1.82\times$, as compared to greedy decoding and random sampling, respectively, when applied to LlamaGen, a contemporary visual AR model. The code is publicly available at https://github.com/jadohu/LANTERN.

## 1 INTRODUCTION

Auto-Regressive (AR) models have recently gained significant traction in image generation (Ramesh et al., 2021; Chen et al., 2020; Tian et al., 2024; Sun et al., 2024) due to their competitive performance, often matching or even surpassing diffusion models (Ho et al., 2020; Rombach et al., 2022). Notable examples include iGPT (Chen et al., 2020), DALL-E (Ramesh et al., 2021), VAR (Tian et al., 2024), and LlamaGen (Sun et al., 2024), which showcase the potential of AR models in image generation. Moreover, recent studies like Lu et al. (2023); Team (2024); Chern et al. (2024) have demonstrated that AR modeling can handle multi-modal data, including language and images, within a single unified framework. Given the remarkable success of AR models in language modeling, leading to the era of large language models (LLMs) (Brown et al., 2020; Touvron et al., 2023; Jiang et al., 2023), it is anticipated that AR modeling will emerge as a dominant paradigm for unifying multiple modalities into a single model in the near future.

Despite the promising potential of AR models, their sequential nature poses a significant bottleneck for both efficiency and scalability since they generate a single token per forward pass. In contrast,

---

[*]Equal Contribution

[†]Corresponding Authors

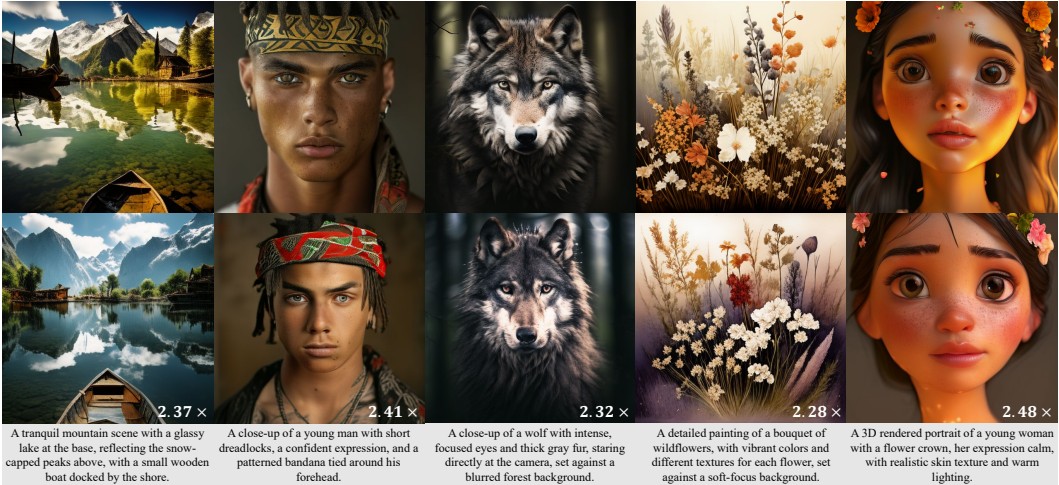

| | | | | |
|---|---|---|---|---|
| A tranquil mountain scene with a glassy lake at the base, reflecting the snow-capped peaks above, with a small wooden boat docked by the shore. | A close-up of a young man with short dreadlocks, a confident expression, and a patterned bandana tied around his forehead. | A close-up of a wolf with intense, focused eyes and thick gray fur, staring directly at the camera, set against a blurred forest background. | A detailed painting of a bouquet of wildflowers, with vibrant colors and different textures for each flower, set against a soft-focus background. | A 3D rendered portrait of a young woman with a flower crown, her expression calm, with realistic skin texture and warm lighting. |

Figure 1: Images generated by vanilla decoding *(top)* and lossy speculative decoding with our relaxed acceptance condition *(bottom)* on the text-conditioned LlamaGen-XL Stage II (Sun et al., 2024). The mean accepted length for each image is displayed in white at the bottom right corner of each image.

GANs (Goodfellow et al., 2014; Karras et al., 2019), which generate images in a single forward pass, naturally avoid this issue, and diffusion models have benefited from extensive research aimed at improving their speed (Song et al., 2022; Sauer et al., 2023; Heek et al., 2024). However, transferring these acceleration techniques to visual AR models is far from straightforward due to fundamental differences in the underlying mechanisms of these models.

One notable acceleration technique for AR models is speculative decoding (Leviathan et al., 2023; Chen et al., 2023; Cai et al., 2024; Li et al., 2024b), which has demonstrated its effectiveness in LLMs. Initially introduced by Leviathan et al. (2023), speculative decoding addresses the sequential bottleneck of AR models by introducing a *draft and verify* mechanism. In this framework, a smaller model (the *drafter*) predicts the next few tokens, which are then verified by the larger target model. If the drafter's predictions are accurate, multiple tokens can be generated from a single forward pass, resulting in a substantial inference speed-up. This method has proven highly effective in accelerating LLM inference, making it a leading option for reducing the latency associated with AR models.

While speculative decoding shows great promise for accelerating AR models, its application to visual AR models remains largely unexplored. Therefore, in this paper, we take the first step toward addressing this gap by migrating speculative decoding to visual AR models. Interestingly, our findings reveal that the naïve application of existing speculative decoding methods falls short in visual AR models. Specifically, we identify a key problem, namely the ***token selection ambiguity***, that hampers the effective migration of speculative decoding to visual AR models.

To mitigate such an obstacle in speculative decoding, we propose a solution dubbed as ***LANTERN*** (**La**tent **N**eighbor **T**oken Acc**e**ptance **R**elaxatio**n**) that leverages the interchangeability of image tokens in latent space for the relaxation of acceptance condition. By relaxing the acceptance in speculative decoding, we allow for more effective utilization of draft (candidate) tokens that would otherwise be frequently rejected despite their potential usefulness. However, our relaxation introduces some distortion to the target model's distribution, which may cause the generated images to deviate from the original target output. To mitigate this, we further incorporate a total variation distance bound which ensures that the deviation remains controlled.

Our main contributions are summarized as below:

- To the best of our knowledge, we are the first to thoroughly investigate speculative decoding in visual AR models, identifying the *token selection ambiguity* problem, where near-uniform token probability distributions hinder token prioritization, causing existing methods to fail in improving speed.

- Based on our insights, we then propose LANTERN, a novel relaxation of acceptance condition for the speculative decoding that addresses the token selection ambiguity problem, successfully enabling the effective application of speculative decoding to visual AR models.

- Our experiments using LlamaGen (Sun et al., 2024) as the target and EAGLE-2 (Li et al., 2024a) as the base speculative decoding method demonstrate significant speed-ups, improving from $1.29\times$ to $\mathbf{2.26\times}$ in greedy decoding and from $0.93\times$ to $\mathbf{1.69\times}$ in random sampling, compared to the naive application of EAGLE-2, without substantial performance drop in terms of image quality.

## 2 PRELIMINARIES

**Notations** In this paper, the *target model* refers to the visual AR model we aim to accelerate. In contrast, the *drafter model* is a supplementary model used to generate draft tokens. The probability distributions modeled by the drafter and target models are represented by $p(\cdot|\cdot)$ and $q(\cdot|\cdot)$, respectively. Individual tokens are denoted in lowercase $x$, and sequences are represented by uppercase $X$; for instance, $X_{i:j}$ represents the sequence $(x_i, \ldots, x_j)$. The concatenation of sequences $X_{1:N}$ and $Y_{1:M}$ is denoted by $(X_{1:N}, Y_{1:M}) = (x_1, \ldots, x_N, y_1, \ldots, y_M)$. For simplicity, we allow certain notational liberties; for instance, expressions like $X_{1:0}$ are treated as empty sequences to avoid unnecessary complexity in notation.

**Visual Autoregressive Modeling** In visual AR models, image generation involves two main stages: generating image tokens through auto-regression and decoding the image tokens into actual image patches. In a text-to-image generation setting, given a tokenized text prompt $X_{1:N}$, the model generates a sequence of image tokens $X_{N+1:N+K}$ based on the following probability modeling:

$$P(X_{N+1:N+K} \mid X_{1:N}) = \prod_{\ell=1}^{K} P(x_{N+\ell} \mid X_{1:N+\ell-1}),$$

where $K$ represents the total number of image tokens corresponding to the height and width of the image feature map. Since each token is predicted based solely on its preceding tokens, visual AR models require $K$ sequential steps to generate all $K$ image tokens.

Once the $K$ image tokens are generated, they are mapped to visual representation by referring to the codebook $\mathcal{C}$. Specifically, the codebook $\mathcal{C} = \{c_1, \ldots, c_L\}$ consists of codes $c_i \in \mathbb{R}^d$, where each code is a latent feature used by the image encoder-decoder pair (such as VQVAE (Van Den Oord et al., 2017) or VQGAN (Esser et al., 2021)), and $d$ is the dimensionality of these features. As the text tokenization, since the image tokens represent indices in the codebook $\mathcal{C}$, each image token $x_{N+i}$ maps to $c_{x_{N+i}}$, which is then rearranged into a $h \times w \times d$ shaped tensor in raster-scan order (from top-left to bottom-right), for $h = H/f$ and $w = W/f$ when $f$ is down-sampling factor. After that, rearranged latent is fed into the image decoder $D : \mathbb{R}^{h \times w \times d} \to \mathbb{R}^{H \times W \times C}$ to construct an actual RGB image.

**Speculative Decoding** We briefly introduce the basics of speculative decoding, which were proposed by Leviathan et al. (2023). Given a sequence of tokens $X_{1:N} = (x_1, \ldots, x_N)$, the drafter model generates $\gamma$ draft tokens $\widetilde{X}_{1:\gamma}$ as speculations for the next $\gamma$ tokens following $X_{1:N}$. Each draft token $\tilde{x}_i$ is sampled from the drafter distribution $p(x|(X_{1:N}, \widetilde{X}_{1:i-1}))$ for each $i = 1, \ldots, \gamma$, thus requiring $\gamma$ forward steps to generate $\gamma$ draft tokens.

After obtaining the draft tokens, the concatenated sequence $(X_{1:N}, \widetilde{X}_{1:\gamma})$ is fed into the target model, which calculates the likelihood of each draft token $q(\tilde{x}_i|(X_{1:N}, \widetilde{X}_{1:i-1})$ in parallel within a single forward pass. Each draft token $\tilde{x}_i$ is accepted with a probability:

$$\min\left(1, \frac{q(\tilde{x}_i|(X_{1:N}, \widetilde{X}_{1:i-1}))}{p(\tilde{x}_i|(X_{1:N}, \widetilde{X}_{1:i-1}))}\right).$$

If $\tilde{x}_i$ is accepted, it is immediately set as $x_{N+i} = \tilde{x}_i$. Otherwise, all subsequent tokens $\widetilde{X}_{i+1:\gamma}$ are discarded, and $x_{N+i}$ is resampled from a distribution defined by $[q(x|(X_{1:N}, \widetilde{X}_{1:i-1})) - p(x|(X_{1:N}, \widetilde{X}_{1:i-1}))]_+$, where $[\cdot]_+$ denotes normalization over positive values only. As proven by Leviathan et al. (2023), this approach ensures that the distribution of the generated token sequence matches the distribution produced by the target model. Related works on the visual AR models and speculative decoding are presented in Appendix A.

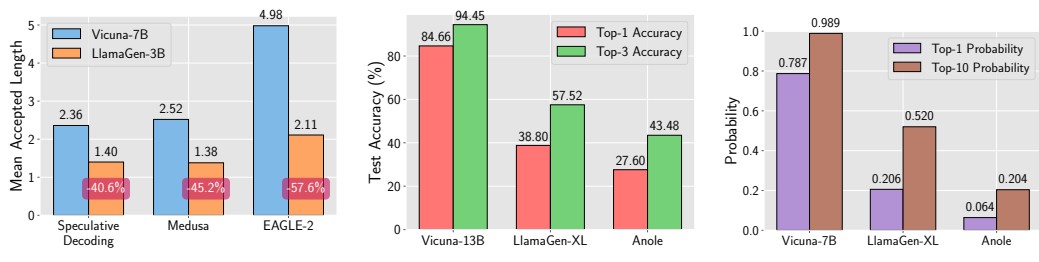

(a) Naïve Application Performance  (b) Drafter Test Accuracy  (c) Average Top-1,10 probabilities

Figure 2: (a) Mean accepted length of naïve application of existing speculative decoding methods on visual AR model and LLM counterpart. (b) Top-1 and top-3 accuracy of learned drafter model for predicting the target model's outputs. (c) An average top-1 and top-10 probabilities in the next token prediction.

## 3 Token Selection Ambiguity Limits Speculative Decoding in Visual AR Models

Although speculative decoding has been highly successful in LLMs, it fails to deliver comparable speed improvements when naïvely applied to visual AR models. As illustrated in Figure 2(a), the visual AR model (LlamaGen-3B (Sun et al., 2024)) exhibits a 41% to 58% reduction in mean accepted length compared to their performance in LLM (Vicuna-7B (Zheng et al., 2023)), a consistent decline observed across different speculative decoding methods. Toward an in-depth exploration of this performance degradation, we train drafters for various visual AR models and conduct additional analyses on their predictions.

Our analysis reveals that the drafter in visual AR models (LlamaGen-XL and Anole (Chern et al., 2024)) struggles to predict the target model's outputs accurately. Specifically, as demonstrated in Figure 2(b), the trained drafters for visual AR models fail to capture the target model's prediction precisely, whereas the drafter in LLMs exhibits considerably higher accuracy. Such a low accuracy of the drafters for visual AR models leads to a significant reduction in the speed-up achievable via speculative decoding.

To delve into the root cause of the drafter's performance degradation, we analyze the next token probabilities of visual AR models and identify a unique problem we term *token selection ambiguity*. As shown in Figure 2(c), visual AR models present substantially lower average top-1 and top-10 probabilities in next token predictive distributions compared to language models, indicating that visual AR models are more ambiguous when selecting the next token. This lack of prioritization among tokens reflects the model's limited confidence in any single option.

We hypothesize that this issue arises from fundamental differences between image and language data. In language models, tokens represent discrete units, such as words or subwords, that form structured and predictable sequences governed by grammar and syntax (Zipf, 1935). Consequently, the next-token probabilities are more concentrated, and the model has high confidence in the most likely token. In contrast, visual AR models treat pixels or patches as tokens, forming continuous and highly complex sequences. As a result, these models exhibit more dispersed next token probabilities and face higher uncertainty and ambiguity in predicting subsequent tokens. This *token selection ambiguity* problem in visual AR models hinders the drafter's predictive accuracy, thereby limiting the effectiveness of speculative decoding. Further details on the setup of experiments can be found in Appendix B.1.

## 4 Detouring Token Selection Ambiguity Through Latent Space

In this section, we propose LANTERN, a simple yet effective method that permits detouring the failure of speculative decoding caused by the token selection ambiguity problem by relaxing acceptance condition in Speculative Decoding (Leviathan et al., 2023). In Section 4.1, we introduce a concept of latent proximity which asserts close image tokens in the latent space are interchangeable and examines its validity on the generated images. Section 4.2 describes how we relax the acceptance

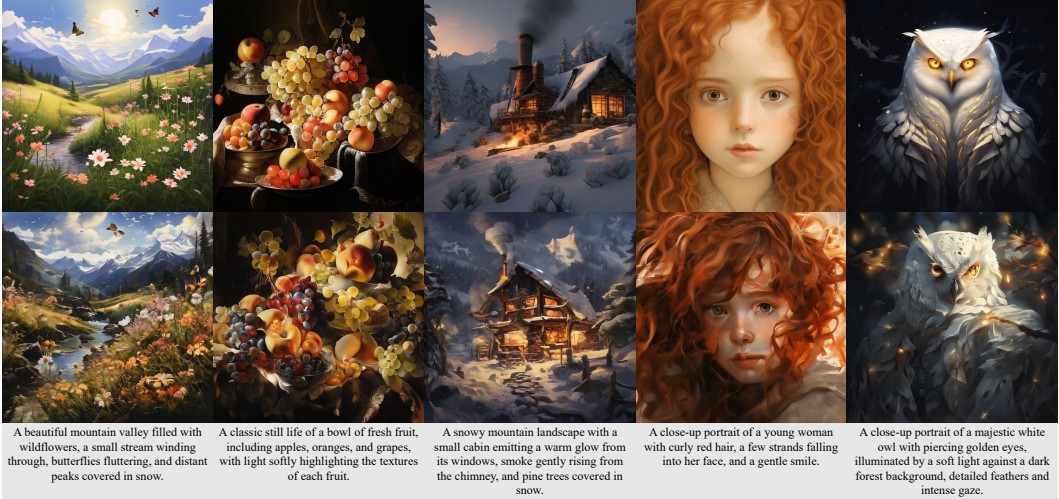

| A beautiful mountain valley filled with wildflowers, a small stream winding through, butterflies fluttering, and distant peaks covered in snow. | A classic still life of a bowl of fresh fruit, including apples, oranges, and grapes, with light softly highlighting the textures of each fruit. | A snowy mountain landscape with a small cabin emitting a warm glow from its windows, smoke gently rising from the chimney, and pine trees covered in snow. | A close-up portrait of a young woman with curly red hair, a few strands falling into her face, and a gentle smile. | A close-up portrait of a majestic white owl with piercing golden eyes, illuminated by a soft light against a dark forest background, detailed feathers and intense gaze. |

Figure 3: Image generated by text-conditioned LlamaGen-XL Stage II model (Sun et al., 2024). The images are generated by either standard sampling method *(top)* or sampling with random replacement within 100-closest tokens in the latent space *(bottom)*.

condition based on the interchangeability. In Section 4.3, we present another component to ensure that the distribution of generated images does not catastrophically deviate from the original distribution.

## 4.1 LATENT PROXIMITY PERMITS TOKEN INTERCHANGEABILITY

We introduce *latent proximity*, a property in visual AR models that asserts tokens close to one another in latent space are *interchangeable* without significantly affecting the visual semantics or overall image quality. This means that replacing a token with another nearby token in latent space results in minimal changes to the generated image.

This property arises from the tokenization process unique to visual AR models. Unlike text tokenization, which is straightforward due to its discrete nature, images are spatially continuous, making tokenization more complex Esser et al. (2021). To handle this, models like VQVAEs (Van Den Oord et al., 2017) and VQGANs (Esser et al., 2021) are used to discretize the latent embeddings of images, as introduced in Section 2. These embeddings maintain a continuous mapping between changes in latent space and the visual semantics of the generated images (Kingma & Welling, 2022; Goodfellow et al., 2014; Karras et al., 2019). As a result, small shifts in latent space lead to minor shifts in the image, supporting the idea that tokens close in the latent space are effectively interchangeable.

To demonstrate this concept empirically, we perform an experiment in which, after each token is sampled, it is re-sampled uniformly from the 100 closest tokens in latent space. Figure 3 reveals that the images generated by this procedure closely resemble those produced using the original sampling method. This confirms that tokens close in latent space can be treated as interchangeable, allowing for flexible token replacement without significantly compromising the visual semantics or overall image quality. A more detailed analysis of latent proximity can be found in Appendix C.

## 4.2 LANTERN: RELAXATION OF ACCEPTANCE CONDITION

Building on our findings about latent proximity, we introduce LANTERN, a simple yet effective solution that leverages the interchangeability of proximate tokens in latent space. By treating neighboring tokens as commutable, LANTERN effectively resolves the token selection ambiguity problem, significantly boosting the acceptance probability of candidate tokens and enabling the successful application of speculative decoding.

We start with revisiting the original acceptance condition from Leviathan et al. (2023), introduced in Section 2. The drafter model samples a draft token $\widetilde{x} \sim p(x|s)$ given a preceding sequence $s = X_{1:N}$. The draft token is accepted with probability $\min\left(1, \frac{q(\widetilde{x}|s)}{p(\widetilde{x}|s)}\right)$ and if rejected, the next

Table 1: Average accept probabilities of LANTERN. We only use accept probability of the first draft token. An average accept probability of EAGLE-2 (Li et al., 2024a) is **0.0402**.

| | Average Accept Probability | | | |
|---|---|---|---|---|
| $k$ | $\delta = 0.05$ | $\delta = 0.1$ | $\delta = 0.2$ | $\delta = 0.4$ |
| 100 | 0.0725 | 0.1096 | 0.1703 | 0.2595 |
| 300 | 0.0759 | 0.1166 | 0.1892 | 0.3267 |
| 1000 | 0.0786 | 0.1186 | 0.2000 | 0.3657 |

token is re-sampled from $[q(x|s) - p(x|s)]_+$. Note that the acceptance depends on the alignment of probabilities between the drafter and target models.

However, this acceptance condition results in a sharp decline in accept probability when encountering the token selection ambiguity problem. As mentioned in Table 1, the EAGLE-2 drafter exhibits an average accept probability of 0.0402, meaning *only 4% of drafts are accepted* when applied to visual AR models. This issue arises because the target model assigns low probabilities to individual tokens and frequently misaligns with the drafter's distribution, leading to frequent rejections of candidate tokens and reducing the overall effectiveness of speculative decoding.

To alleviate this problem, we exploit the latent proximity by aggregating the probabilities of a candidate token's nearest neighbors, treating them as proxies. This approach effectively increases the acceptance probability by utilizing the combined likelihood of similar tokens, mitigating the impact of the token selection ambiguity problem and reducing unnecessary rejections.

Specifically, we define the neighborhood $B_k(\widetilde{x})$ as the set of $k$-nearest tokens to $\widetilde{x}$ in latent space, including $\widetilde{x}$ itself. The accept probability is then adjusted to

$$\min \left( 1, \frac{\sum_{x \in B_k(\widetilde{x})} q(x|s)}{p(\widetilde{x}|s)} \right). \tag{1}$$

Because $\widetilde{x}$ is always included in $B_k(\widetilde{x})$, this new accept probability is guaranteed to be equal to or higher than the original acceptance condition. As demonstrated in Table 1, applying LANTERN significantly increases the average acceptance probability, reaching values as high as 0.37. This improvement allows us to recover candidate tokens that would have otherwise been unjustifiably rejected.

### 4.3 WITH LIMITED DISTRIBUTIONAL DIVERGENCE

Although the relaxed acceptance condition (1) effectively permits speculative decoding in visual AR models by significantly raising the accept probability, it inevitably distorts the target distribution. In particular, when we condition the target distribution on the candidate token $\widetilde{x}$, it becomes:

$$q_k(x|s, D = \widetilde{x}) = \begin{cases} \sum_{x \in B_k(\widetilde{x})} q(x|s) & \text{if } x = \widetilde{x} \\ 0 & \text{if } x \in B_k(\widetilde{x}), x \neq \widetilde{x} \\ q(x|s) & \text{otherwise} \end{cases}$$

where $D$ is a random variable representing the candidate token and $q_k$ denotes the distorted target distribution. In contrast, under the original acceptance condition, the target distribution remains unchanged regardless of the candidate token. For this reason, (1) may excessively distort the target distribution, leading to generating images that diverge significantly from those generated by the target model.

To mitigate this distortion, we impose an upper bound on the distributional divergence using total variation distance (TVD). Since the distortion results from redistributing probability mass, TVD effectively measures the extent of this shift, allowing us to control the magnitude of the divergence. This can be achieved by adjusting the neighborhood $B_k(\widetilde{x})$ used in the relaxation as follows.

Since the relaxation can be analogously derived using any neighborhood of $\widetilde{x}$, we can find a neighborhood that ensures the TVD between the target distribution and the distorted target distribution induced by the neighborhood is below a specific threshold. To formulate this approach, we define

the neighborhood $A_{k,\delta}(\widetilde{x})$ of $\widetilde{x}$ for a given TVD bound $\delta > 0$ and $k \in \mathbb{Z}^+$ as $A_{k,\delta}(\widetilde{x})$ is the largest subset of $B_k(\widetilde{x})$ such that for the total variation distance $D_{TV}$,

$$D_{TV}\left(q_{k,\delta}(x|s, D = \widetilde{x}), q(x|s, D = \widetilde{x})\right) = D_{TV}(q_{k,\delta}(x|s, D = \widetilde{x}), q(x|s)) < \delta$$

where $q_{k,\delta}$ denotes the distorted target distribution induced by $A_{k,\delta}(\widetilde{x})$.

We construct $A_{k,\delta}(\widetilde{x})$ by incrementally adding tokens from $B_k(\widetilde{x})$ to $A_{k,\delta}(\widetilde{x})$, starting with the closest ones to $\widetilde{x}$, and stopping when adding another token would exceed the TVD threshold $\delta$. This procedure allows us to relax the acceptance condition by incorporating probabilities of similar tokens while keeping the divergence within a predefined boundary.

By integrating the TVD constraint into the acceptance condition (1), we arrive at the final relaxed acceptance condition of LANTERN:

$$\textbf{Accept } \widetilde{x} \text{ with probability } \min\left(1, \frac{\sum_{x \in A_{k,\delta}(\widetilde{x})} q(x|s)}{p(\widetilde{x}|s)}\right)$$

$$\textbf{Else } \text{re-sample } x \sim [q_{k,\delta}(x|s, D = \widetilde{x}) - p(x|s)]_+$$

For greedy decoding, LANTERN can be simply reduced to accept $\widetilde{x}$ if $\widetilde{x} = \arg\max_x q_{k,\delta}(x|s, D = \widetilde{x})$. The algorithms for LANTERN and construction of $A_{k,\delta}$ can be found in Appendix D.

## 5 EXPERIMENTS

### 5.1 EXPERIMENTAL SETUP

To validate our method LANTERN, we conduct experiments on text-conditioned LlamaGen-XL Stage I model (Sun et al., 2024) as a target model that establishes the best performance among visual AR models without vision-specific modifications. We utilize the MS-COCO validation captions (Lin et al., 2014) to generate images and evaluate the image quality with the ground-truth images. For the base speculative decoding method, we employ EAGLE-2 (Li et al., 2024a), which has demonstrated state-of-the-art performance in speculative decoding in the language domain. We employ the $\ell_2$ distance to quantify latent proximity and utilize the TVD as a metric for divergence bound.

For the assessment of speed-ups, we use 1000 MS-COCO validation captions because this sample size provides results that are nearly identical to those obtained when evaluating the entire dataset. Actual speed-up is measured by the inference time ratio between each method and standard auto-regressive decoding. The mean accepted length is determined by the average number of tokens accepted in each forward step of the target model. We evaluate each method in both the greedy decoding setting with $\tau = 0$ and the sampling with $\tau = 1$. The statistical analysis on actual speed-up and number of captions can be found in Appendix F.1.

Since LANTERN can impact the quality of generated images, we evaluate image quality using FID (Heusel et al., 2017), CLIP score (Hessel et al., 2021), Precision and Recall (Kynkäänniemi et al., 2019), and HPS v2 (Wu et al., 2023) with 30K samples. For each evaluation, we do not evaluate image quality on EAGLE-2 since it theoretically guarantees exact distribution matching with the target model. Further details can be found in Appendix B.2.

### 5.2 MAIN RESULTS

In this section, we demonstrate qualitative and quantitative results of speculative decoding with our relaxed acceptance condition. First of all, Section 5.2.1 demonstrates how much speed-up can be achieved through our method. Next, Section 5.2.2 showcases that our method retains image quality within a similar level by showing qualitative samples. Afterward, Section 5.2.3 presents the trade-off between performance and efficiency in our method.

### 5.2.1 SPEED UP COMPARISON

To confirm that LANTERN provides notable speed improvements over the baseline while maintaining image quality, we compare our method with baselines under both greedy decoding and random

Table 2: Actual speed-up, MAL (Mean Accepted Length), FID, CLIP score, Precision / Recall, and HPS v2 for each method. $\tau = 0$ refers to the greedy decoding and $\tau = 1$ refers to the sampling with temperature 1 for generation. The actual speed-up is measured on a single RTX 3090.

| Method | $\tau = 0$ | | | | | | |
| | Acceleration (↑) | | Image Quality Metrics | | | | |
| | Speed-up | MAL | FID (↓) | CLIP score (↑) | Precision (↑) | Recall (↑) | HPSv2 (↑) |
|---|---|---|---|---|---|---|---|
| Vanilla AR (Sun et al., 2024) | 1.00× | 1.00 | 28.63 | 0.3169 | 0.4232 | 0.3517 | 23.18 |
| EAGLE-2 (Li et al., 2024a) | 1.29× | 1.60 | - | - | - | - | - |
| LANTERN ($\delta = 0.05, k = 1000$) | 1.56× | 2.02 | 29.77 | 0.3164 | 0.4484 | 0.3158 | 22.62 |
| LANTERN ($\delta = 0.2, k = 1000$) | **2.26×** | **2.89** | 30.78 | 0.3154 | 0.4771 | 0.2773 | 21.69 |
| Method | $\tau = 1$ | | | | | | |
| | Acceleration (↑) | | Image Quality Metrics | | | | |
| | Speed-up | MAL | FID (↓) | CLIP score (↑) | Precision (↑) | Recall (↑) | HPSv2 (↑) |
| Vanilla AR (Sun et al., 2024) | 1.00× | 1.00 | 15.22 | 0.3203 | 0.4781 | 0.5633 | 24.11 |
| EAGLE-2 (Li et al., 2024a) | 0.93× | 1.20 | - | - | - | - | - |
| LANTERN ($\delta = 0.1, k = 1000$) | 1.13× | 1.75 | 16.17 | 0.3208 | 0.4869 | 0.5172 | 23.75 |
| LANTERN ($\delta = 0.4, k = 1000$) | **1.69×** | **2.40** | 18.76 | 0.3206 | 0.4909 | 0.4497 | 23.22 |

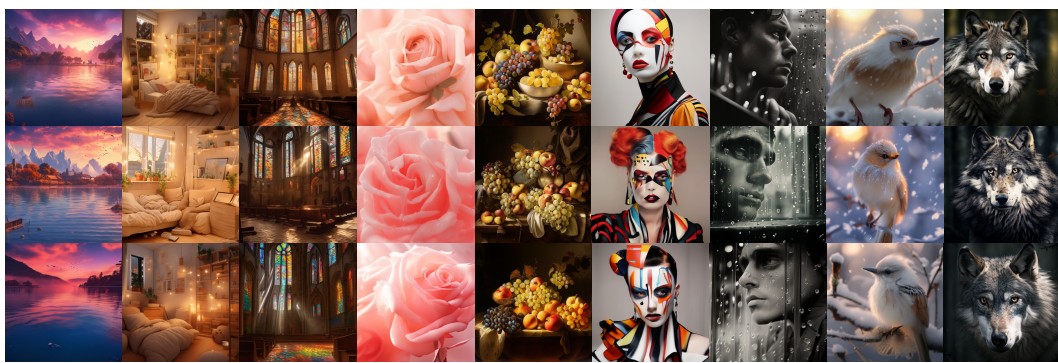

Figure 4: Qualitative samples generated by LlamaGen-XL Stage II model for LANTERN and standard autoregressive decoding. From top to bottom, the images are generated by standard autoregressive decoding, LANTERN ($\delta = 0.2, \delta = 0.4$) where $k$ is fixed at 1000, and images in the same column are generated using the same text prompt. Text prompts for the images are provided in Appendix I.

sampling situations. Table 2 demonstrates the speed-up and image quality across different methods. LANTERN shows a significant acceleration, even with a slight degradation in image quality, when compared to standard autoregressive decoding and EAGLE-2 (Li et al., 2024a). Our method, LANTERN demonstrates significant improvements in both mean accepted length and actual speed-up. In terms of mean accepted length, LANTERN outperforms both baselines by reaching 2.40× and it translate to substantial actual speed-up in practice, with LANTERN achieving 1.69× actual speed-up.

While LANTERN's acceleration comes with a trade-off in image quality, the degradation remains minimal and well within acceptable boundaries. For instance, the HPS v2 score decreases by less than 1.5 for both greedy decoding and sampling when compared to standard autoregressive decoding. Precision remains stable or slightly improved, whereas recall shows a modest decline, indicating a slight reduction in image diversity but with individual image quality largely preserved. Other metrics, such as FID and CLIP score, further confirm that LANTERN maintains competitive image quality despite the significant acceleration.

As a result, by allowing a degree of flexibility in token selection, our method strikes a favorable balance between speed and quality, outperforming both standard autoregressive decoding and EAGLE-2 in terms of practical efficiency. Experimental results on other visual AR models including LlamaGen-XL Stage II model and Anole (Chern et al., 2024) are presented in Appendix E. Also, detailed analyses of latency, including latency on Intel Gaudi 2, are provided in Appendix F.

### 5.2.2 QUALITATIVE RESULTS

To confirm that image quality is preserved with LANTERN, as indicated by the various image quality metrics, we conduct a qualitative analysis. Figure 4 demonstrates that, despite the modification of

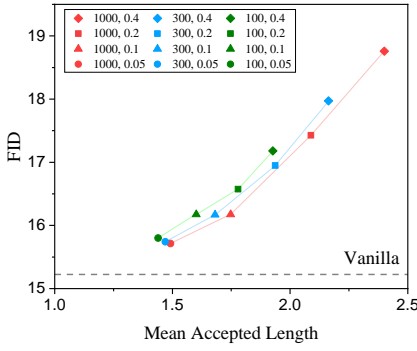 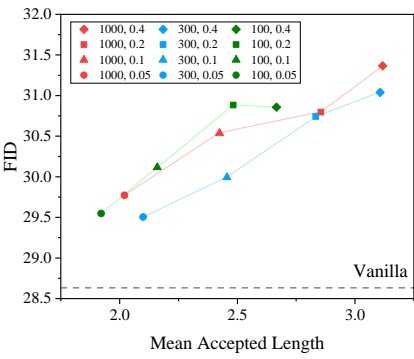

(a) Trade-off curve with $\tau = 1$ (sampling)       (b) Trade-off curve with $\tau = 0$ (greedy)

Figure 5: Trade-off curves show the relationship between performance (FID) and acceleration (mean accepted length). The results with the same $k$ are annotated with the same color, while the same $\delta$ values are marked with identical symbols. In the legend, the values are separated by commas, indicating $k$ and $\delta$, respectively.

the target model's probability distribution to achieve acceleration, our method effectively preserves image quality. Notably, even under the setting of $\delta = 0.4$ and $k = 1000$, which achieve about $1.64\times$ speed-up compared to the standard autoregressive decoding, generated images retain both content and style at a level comparable to standard autoregressive decoding. These qualitative results, along with the fact that LANTERN avoids significant degradation in image quality metrics as shown in the previous section, demonstrate that it effectively preserves image quality while increasing efficiency. More qualitative examples can be found in Appendix J.

### 5.2.3 TRADE-OFF BETWEEN PERFORMANCE AND EFFICIENCY

LANTERN provides various options between quality and efficiency to the end users. Therefore, we explore this trade-off across different hyperparameters by adjusting $k$ and $\delta$. Figure 5 illustrates the relationship between image quality, which is measured by FID, and speed-up, which is assessed with mean accepted length for various settings of $\delta$ and $k$, under both sampling ($\tau = 1$) and greedy decoding ($\tau = 0$). The trade-off curves highlight that increasing $\delta$ and $k$ generally improves speed-up, but at the expense of image quality.

The results underline that LANTERN allows for flexible tuning to balance acceleration and image quality depending on specific practical requirements. For instance, in the sampling ($\tau = 1$), configurations with smaller $\delta$ or $k$ prioritize image quality, maintaining lower FID scores, while larger $\delta$ or $k$ achieve greater acceleration with a controlled loss in quality. A similar trend is observed for greedy decoding ($\tau = 0$), where the trade-off is more pronounced at higher mean accepted lengths. Importantly, across all hyperparameter configurations, LANTERN demonstrates consistent and predictable trade-offs, enabling users to fine-tune the method based on their preferred balance between speed and quality. Trade-offs with other image quality metrics can be found in Appendix H.

### 5.3 ABLATION STUDY

In this section, we conduct ablation studies for LANTERN. In Section 5.3.1, we assess the impact of the metric used to measure latent proximity on performance. Then, in Section 5.3.2, we provide an ablation study on the effect of the metric for measuring the distance from the modified probability distribution.

### 5.3.1 NEAREST LATENT SELECTION

To explore the impact of various metrics for measuring latent proximity, we conduct an ablation study using representative distance metrics commonly used to measure latent proximity. Table 3 summarizes the comparisons among different strategies for selecting the nearest latent tokens, including $\ell_2$ distance, cosine similarity, and random selection under sampling ($\tau = 1$). This experiment aims to assess the role of proximity-based selection in token aggregation, with $k = 1000$ used across all methods.

Table 3: Ablation study for latent proximity measure and probability distribution distance metrics on LlamaGen-XL Stage I model under sampling ($\tau = 1$). For latent proximity measures, $\ell_2$ distance, cosine similarity, and random selection are used, and TVD and JSD are used as probability distribution distance metrics.

| Distance Metric | Latent Proximity Measure | | |
|---|---|---|---|
| | Mean Accepted Length | FID | CLIP score |
| Cosine similarity ($\delta = 0.2$) | 2.09 | 17.46 | 0.3206 |
| $\ell_2$ distance ($\delta = 0.2$) | 2.09 | 17.43 | 0.3208 |
| $\ell_2$ distance ($\delta = 0.05$) | 1.50 | 15.71 | 0.3203 |
| Random ($\delta = 0.2$) | 1.26 | 15.62 | 0.3203 |
| Distance Metric | Probability Distribution Distance | | |
| | Mean Accepted Length | FID | CLIP score |
| TVD ($\delta = 0.3$) | 2.29 | 18.27 | 0.3206 |
| JSD ($\delta = 0.2$) | 2.29 | 18.21 | 0.3206 |
| TVD ($\delta = 0.2$) | 2.09 | 17.43 | 0.3208 |
| JSD ($\delta = 0.13$) | 2.09 | 17.48 | 0.3206 |

As shown in Table 3, the random selection significantly underperforms in terms of acceleration, achieving only a 1.26 mean accepted length, which is notably lower than both $\ell_2$ distance and cosine similarity. Additionally, the random selection shows inferior acceleration compared to the $\ell_2$ distance with $\delta = 0.05$, while maintaining a similar FID, which highlights the importance of token selection based on latent proximity. Comparing $\ell_2$ distance and cosine similarity, both methods demonstrate nearly identical performance in terms of mean accepted length, FID, and CLIP score, suggesting robustness and flexibility in our approach to proximity measurement. These results emphasize that selecting tokens based on latent proximity is critical for achieving a balance between acceleration and image quality, offering a clear advantage over random selection. An analysis on the size of latent proximity set is presented in Appendix G.

### 5.3.2 DISTANCE BETWEEN PROBABILITY DISTRIBUTION

To ensure that the modified target distribution remains within an acceptable range of divergence from the original, we introduce $\delta$ as an upper bound for distributional divergence. To validate the impact of the divergence metric, we evaluate two different metrics to measure the divergence: Total Variation Distance (TVD) and Jensen-Shannon Divergence (JSD). Kullback-Leibler Divergence (KLD) is not used as it is asymmetric and not a valid metric for distance in a mathematical sense.

To compare the effectiveness of these distance metrics, we fix $k = 1000$ and adjust $\delta$ to achieve similar mean accepted lengths across the different divergence metrics. The results shown in Table 3 demonstrate that, although the two metrics require different $\delta$ for the same level of acceleration (measured by mean accepted length), they produce nearly identical image quality metrics at comparable acceleration levels.

These results confirm that our method consistently functions as a robust trade-off controller regardless of the chosen distance metric. Since the difference in performance between two metrics is marginal, we opt to use TVD, as it is computationally lighter and thus more efficient for large-scale implementations. The detailed analysis on the latency differences between TVD and JSD is provided in Appendix F.3.

## 6 CONCLUSIONS

In this paper, we explored the application of speculative decoding to visual AR models for the first time. We revealed that the naïve application of existing methods fails due to the token selection ambiguity problem. To address this, we proposed LANTERN, a novel relaxed acceptance condition that effectively resolves this problem. Our experiments using the state-of-the-art visual AR model and speculative decoding method demonstrated that LANTERN successfully enables speculative decoding in visual AR models, achieving substantial speed-ups with minimal compromise in image generation performance. For future work, we plan to design a drafter specifically tailored to visual AR models, aiming to achieve acceleration without sacrificing the generation performance.

ACKNOWLEDGMENTS

This work was partly supported by Institute for Information & communications Technology Promotion(IITP) (No.RS-2019-II190075, Artificial Intelligence Graduate School Program(KAIST), No.2022-0-00713, Meta-learning applicable to real-world problems, No.RS-2024-00457882, AI Research Hub Project) and National Research Foundation of Korea (NRF) (No.RS-2023- 00209060, A Study on Optimization and Network Interpretation Method for Large-Scale Machine Learning) grant funded by the Korea government (MSIT). This research was supported in part by the NAVER-Intel Co-Lab. The work was conducted by KAIST and reviewed by both NAVER and Intel.

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

# APPENDIX

## A    RELATED WORKS

**Visual auto-regressive models**    The development of auto-regressive (AR) models in language modeling (Brown et al., 2020; Touvron et al., 2023; Jiang et al., 2023) has paved the way for AR-based approaches in generative vision tasks, where images are tokenized into discrete tokens on a 2D grid and processed as a unidirectional sequence. Early studies (Esser et al., 2021; Yu et al., 2021; Lee et al., 2022) explored different sequence orders such as row-major raster scans, spirals, and z-curves for image generation. However, these approaches often suffer from computational inefficiencies and underperform compared to diffusion models (Ho et al., 2020; Song et al., 2022; Rombach et al., 2022).

Building upon fundamental mechanism of visual AR models from DALL-E (Ramesh et al., 2021) and CogView (Ding et al., 2021), where images are discretized into tokens using vector-quantized VAEs (VQVAE) (Van Den Oord et al., 2017) or VQGANs (Esser et al., 2021) for next-token prediction, recently, visual AR models have emerged as strong competitors to diffusion models. For instance, Parti (Yu et al., 2022) and LlamaGen (Sun et al., 2024) utilize a verbose encoder-decoder architecture that incorporates frozen T5 (Raffel et al., 2020) text features via cross-attention or prefix-filling methods, drawing on insights from Imagen (Saharia et al., 2022). More recently, Chameleon (Team, 2024) has proposed a unified AR framework that enables fully token-based representations for both image and text modalities. Additionally, model like Anole (Chern et al., 2024), which build on Chameleon, further enhance image generation capabilities, demonstrating versatility across diverse tasks. In this work, we conduct experiments on state-of-the-art visual AR models—LlamaGen and Anole—for text-to-image generation to validate the effectiveness of our acceleration scheme.

**Speculative decoding for AR models**    The foundational concept of speculative decoding was first introduced by Leviathan et al. (2023) to address the sequential constraints of the AR framework in language modeling. Unlike the standard AR modeling, which generates one token per forward pass, speculative decoding aims to generate multiple tokens in a single forward pass by speculating a series of tokens, referred to as *draft* tokens. To generate draft tokens efficiently, most speculative decoding methods rely on a separate, typically smaller, model called the *drafter* model. Since generating draft tokens with the drafter model is much faster than the target model, this approach can reduce the overall token generation time.

In Leviathan et al. (2023), the drafter model is a smaller version within the same architecture family, trained on similar data and objectives as the target model. An alternative approach, DistillSpec (Zhou et al.), uses knowledge distillation to develop efficient drafter models. Recognizing that the latency of generating draft tokens is crucial for accelerating, recent works have explored lightweight designs for drafter models. Medusa (Cai et al., 2024), for instance, employs multiple separate language model heads as drafter models instead of fully independent models and leverages tree drafting and decoding to enhance the chance of accepting draft tokens. Hydra (Ankner et al., 2024) builds on Medusa by incorporating sequential dependency, yielding additional acceleration benefits. EAGLE (Li et al., 2024b;a) extends these approaches by incorporating feature uncertainty in drafting and utilizing a single decoder layer as drafter models to incorporate better sequential dependency. Furthermore, they have achieved state-of-the-art performance in speculative decoding for language modeling by dynamically refining the tree structure for tree drafting.

# B EXPERIMENTAL DETAILS

## B.1 EXPERIMENTS FOR MOTIVATING EXAMPLES

For the experiment on naïve application shown in Figure 2(a), we utilize LlamaGen-L (Sun et al., 2024), an MLP with two linear layers, and a single decoder layer as the drafters for Speculative Decoding (Leviathan et al., 2023), Medusa (Cai et al., 2024), and EAGLE-2 (Li et al., 2024a), respectively. To facilitate Speculative Decoding, which requires a smaller-sized model with the same architecture as the main model and trained on the same dataset, we employ a class-conditioned LlamaGen-3B model instead of text-conditioned models. This is because text-conditioned models do not provide multiple model sizes. For each ImageNet class (Deng et al., 2009), we generate 100 images to measure the mean accepted length. Results on Vicuna-7B (Zheng et al., 2023) are taken directly from EAGLE-2.

The drafter test accuracies presented in Figure 2(b) correspond to the test accuracies of the drafters used in our main experiments (Table 2). Details on the training process for these drafters are provided in the following section. Additionally, the test accuracies of the learned drafter for Vicuna-13B are sourced from EAGLE-2.

For the experiments on average top-1 and top-10 probabilities shown in Figure 2(c), we generate 80 responses from Vicuna-7B on the MT-Bench dataset (Zheng et al., 2023) and 1000 images from LlamaGen-XL and Anole based on MS-COCO validation captions (Lin et al., 2014).

## B.2 EXPERIMENTS FOR MAIN RESULTS

To train the text-conditional model's drafter, we sampled 100k images in LAION-COCO dataset (Chuhmann et al., 2022), which is used to train Stage I target model. We used the same amount of image sampled in ImageNet (Deng et al., 2009) dataset to train the class-conditional model's drafter. For Anole (Chern et al., 2024), we use 118K images sampled with target model by MS-COCO (Lin et al., 2014) train caption to train drafter. We used a single-layer decoder with the same structure as the target model in the same manner as EAGLE. During training, 5% of data is set to be held out validation dataset.

Since LlamaGen (Sun et al., 2024) and Anole (Chern et al., 2024) use classifier-free guidance (Ho & Salimans, 2021) to generate images, we trained our drafter to both learn conditioned input and null-conditioned input. To do so, we dropped 10% of conditional embedding during training, as same as target model training. The batch size is 16, and the base learning rate is $10^{-4}$. AdamW (Loshchilov & Hutter, 2019) optimizer with $\beta_1 = 0.9$ and $\beta_2 = 0.95$ is used, and Linear learning rate scheduling with warm-up is used with 2000 warm-up steps. We select the best-performing model in terms of top-3 accuracy in the hold-out validation set for 20 epochs. In addition, Flan-T5 XL (Chung et al., 2022) is used to encode input text for text-conditional generation.

For LlamaGen (Sun et al., 2024) stage I and stage II, images are generated using a classifier-free guidance scale of 7.5 with top-p set to 1.0 and top-k set to 1000, which is the default generation configuration of LlamaGen official implementation for text-conditional image generation. For a class-conditional generation, the classifier-free guidance scale is set to 4.0, with the top-k sampling covering the entire vocabulary and the top-p sampling set to 1.0. For Anole (Chern et al., 2024), we use a classifier-free guidance scale of 3.0 with with top-k as 2000. For EAGLE-2 and our method, 60 candidate tokens are passed into the target model for each verification process.

## C    DETAILED ANALYSIS ON LATENT PROXIMITY

### C.1    STATISTICAL ANALYSIS ON LATENT PROXIMITY

Table 4: Impact of replacing tokens with one of the $k$-th nearest tokens on image quality. FID and CLIP Score indicate degradation in image quality as $k$ increases.

| Randomly Replaced by one of $k$-th nearest token | FID | CLIP Score |
|---|---|---|
| Vanilla AR | 25.06 | 0.3214 |
| $k = 50$ | 26.88 | 0.3120 |
| $k = 100$ | 30.76 | 0.3091 |
| $k = 1000$ | 88.03 | 0.2715 |

Table 4 provides statistical evidence supporting our earlier qualitative observations (Figure 3) that token replacement does not lead to significant degradation in image quality, particularly for smaller values of $k$. This experiment was conducted using the LlamaGen-XL Stage I model (Sun et al., 2024) and MS-COCO 2017 validation captions (Lin et al., 2014), with FID and CLIP Score used as evaluation metrics.

As $k$ increases, the replaced token is chosen from a larger set of latent space neighbors, and its impact on image quality becomes more evident. For $k = 50$, the FID increases slightly from 25.06 (Vanilla AR) to 26.88, and the CLIP Score decreases marginally from 0.3214 to 0.3120, indicating minimal degradation. Similarly, for $k = 100$, the FID increases to 30.76, and the CLIP Score drops to 0.3091, showing that even with $k = 100$, the image quality remains relatively stable and acceptable. However, for $k = 1000$, a substantial decline is observed, with the FID increasing sharply to 88.03 and the CLIP Score dropping to 0.2715, underscoring the negative impact of replacing tokens with more distant neighbors.

These results corroborate our earlier qualitative findings, showing that token replacement up to $k = 100$ maintains reasonable image quality, making it a viable approach in generative tasks. This demonstrates the robustness of the model under controlled token replacement scenarios.

### C.2    COMPARISON OF LATENT PROXIMITY IN VISUAL AND LANGUAGE AR MODELS

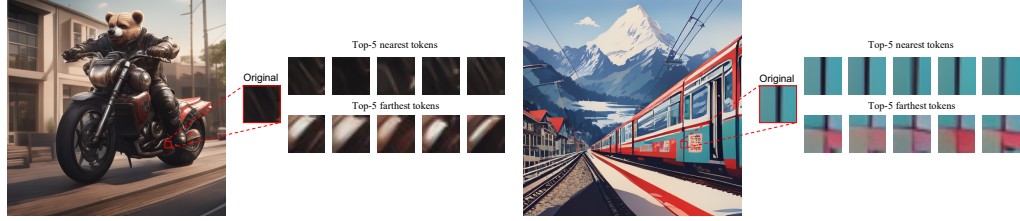

(a) A big teddy bear on a black motorcycle.          (b) A red and white train traveling along a mountain road.

Figure 6: Visualization of top-5 nearest token and top-5 farthest token from original token in LlamaGen-XL Stage II model (Sun et al., 2024).

To achieve a deeper understanding of latent proximity, we conducted additional analyses to examine the tendencies of nearest tokens in visual AR models and language models using LlamaGen-XL Stage II (Sun et al., 2024) and Vicuna-7B (Zheng et al., 2023). For the visual AR model, we visualized the five nearest tokens and the five farthest tokens based on their L2 distances in the latent space, as shown in Figure 6. However, for the language model, since each token corresponds to subwords, it is challenging to measure proximity directly using the token itself. Therefore, we conducted the same analysis using the $\ell_2$ distance of the input embeddings, with the results presented in Table 5.

Table 5: Top-5 nearest and farthest tokens from the original tokens based on $\ell_2$ distance of input embeddings in Vicuna-7B (Zheng et al., 2023). [OBJ] indicates the Unicode object replacement character.

| Top-$k$ | Token: `hi` | | Token: `act` | |
| --- | --- | --- | --- | --- |
| | **Nearest** | **Farthest** | **Nearest** | **Farthest** |
| **1** | _Portály | \n | _Mediabestanden | [OBJ] |
| **2** | _Mediabestanden | [OBJ] | oreferrer | _Bruno |
| **3** | <0x4C> | _infinitely | <0x24> | _Ernst |
| **4** | <0x6B> | _firewall | <0x71> | _Santos |
| **5** | <0x49> | _sooner | <0x54> | _firewall |

This result indicates that in the visual AR model, tokens with high latent proximity are decoded into image patches with similar visual appearances, whereas tokens with low latent proximity correspond to image patches with distinctly different appearances. On the other hand, in the language model, the similarity based on the input embeddings did not reveal a clear relationship between the nearest and farthest tokens. These results support the conclusion that latent proximity serves as an effective metric for identifying visually similar tokens in visual AR models.

# D ALGORITHMS

## D.1 SPECULATIVE DECODING WITH LANTERN

---

**Algorithm 1** LANTERN

---

1: **Input:** Target model $q(\cdot|\cdot)$, draft model $p(\cdot|\cdot)$, initial sequence $x_0, \ldots, x_t$, drafted sequence length $L$, minimum target sequence length $T$, $D_{TV}$ tolerance $\delta > 0$, and maximum cardinality of latent neighborhood $k$.
2: **Initialize:** $n \leftarrow t$.
3: **while** $n < T$ **do**
4:     **for** $t = 1, \ldots, L$ **do**
5:         Sample draft autoregressively $\widetilde{x}_t \sim p(x|x_0, \ldots, x_n, \widetilde{x}_1, \ldots, \widetilde{x}_{t-1})$
6:     **end for**
7:     In parallel, compute $L + 1$ sets of logits from drafts $\widetilde{x}_1, \ldots, \widetilde{x}_L$:

$$q(x|x_0, \ldots, x_n), q(x|x_0, \ldots, x_n, \widetilde{x}_1), \ldots, q(x|x_0, \ldots, x_n, \widetilde{x}_1, \ldots, \widetilde{x}_L)$$

8:     **for** $t = 1, \ldots, L$ **do**
9:         Find the neighborhood $A_{k,\delta}(\widetilde{x}_t)$.
10:         Sample $r \sim U[0, 1]$ from an uniform distribution.
11:         **if** $r < \min\left(1, \frac{\sum_{x \in A_{k,\delta}(\widetilde{x}_t)} q(x|x_0, \ldots, x_{n+t-1})}{p(\widetilde{x}_t|x_0, \ldots, x_{n+t-1})}\right)$ **then**
12:             Set $x_{n+t} \leftarrow \widetilde{x}_t$ and $n \leftarrow n + 1$.
13:         **else**
14:             Sample $x_{n+t} \sim (q_{k,\delta}(x|x_0, \ldots, x_{n+t-1}, D = \tilde{x}_t) - p(x|x_0, \ldots, x_{n+t-1}))_+$ and exit the loop
15:         **end if**
16:     **end for**
17:     If all drafts are accepted, sample an extra token $x_{n+L+1} \sim q(x|x_0, \ldots, x_{n+L})$.
18: **end while**
19: **Output:** $x_{n+1}, \ldots, x_{n+L}$ or $x_{n+1}, \ldots, x_{n+L+1}$

---

## D.2 PROXIMITY SET CONSTRUCTION

---

**Algorithm 2** Proximity Set Construction for LANTERN

---

**Require:** Latent space representation of tokens, number of neighbors $k$, $D_{TV}$ tolerance $\delta$
1: **Precompute $B_k(\widetilde{x})$ for all tokens:**
2: **for** each token $\widetilde{x}$ in the quantized latent space **do**
3:     Compute distances between $\widetilde{x}$ and all other tokens.
4:     Identify $k$ nearest tokens (including $\widetilde{x}$ itself) based on $\ell_2$ distance.
5:     Store these tokens as the set $B_k(\widetilde{x})$.
6: **end for**
7: **Dynamically calculate $A_{k,\delta}(\widetilde{x})$ during inference:**
8: **for** each token $\widetilde{x}$ sampled during decoding **do**
9:     Initialize $A_{k,\delta}(\widetilde{x}) \leftarrow \emptyset$.
10:     Initialize cumulative TVD, $D_{TV} \leftarrow 0$.
11:     **for** each token $x \in B_k(\widetilde{x})$ (in increasing distance order) **do**
12:         Compute potential TVD increment $\Delta D_{TV}$ for adding $x$ to $A_{k,\delta}(\widetilde{x})$.
13:         **if** $D_{TV} + \Delta D_{TV} < \delta$ **then**
14:             Add $x$ to $A_{k,\delta}(\widetilde{x})$.
15:             Update $D_{TV} \leftarrow D_{TV} + \Delta D_{TV}$.
16:         **else**
17:             Break.
18:         **end if**
19:     **end for**
20: **end for**
21: **Return** proximity sets $B_k$ (precomputed) and $A_{k,\delta}$ (dynamically calculated).

---

# E EXPERIMENTAL RESULTS ON OTHER VISUAL AR MODELS

Table 6: Actual speed-up, MAL (Mean Accepted Length), FID, CLIP score, Precision / Recall, and HPS v2 for each method on LlamaGen-XL Stage II and Anole.

| Method | Acceleration (↑) | | Image Quality Metrics | | | | |
| --- | --- | --- | --- | --- | --- | --- | --- |
| | Speed-up | MAL | FID (↓) | CLIP score (↑) | Prec (↑) | Rec (↑) | HPS v2 (↑) |
| LlamaGen-XL Stage II | 1.00× | 1.00 | 47.60 | 0.2939 | 0.4138 | 0.5648 | 23.84 |
| EAGLE-2 (Li et al., 2024a) | 0.96× | 1.22 | - | - | - | - | - |
| LANTERN ($\delta = 0.4$, $k = 1000$) | **1.64×** | **2.24** | 46.10 | 0.2925 | 0.4704 | 0.5222 | 23.06 |
| Anole | 1.00× | 1.00 | 20.27 | 0.3215 | 0.6552 | 0.6398 | 23.52 |
| EAGLE-2 | 0.73× | 1.10 | - | - | - | - | - |
| LANTERN ($\delta = 0.5$, $k = 100$) | **1.17×** | **1.83** | 23.40 | 0.3186 | 0.6026 | 0.6178 | 22.92 |

Table 6 presents additional results comparing LANTERN to EAGLE-2 and standard decoding on LlamaGen-XL Stage II (Sun et al., 2024) and Anole (Chern et al., 2024). The table reports both actual speed-up (measured on RTX 3090 for LlamaGen and A100 80GB SXM for Anole) and image quality metrics, including FID, CLIP score, Precision/Recall, and HPS v2.

As shown in the Table 6, LANTERN consistently achieves superior acceleration compared to EAGLE-2 across multiple visual AR models, including LlamaGen-XL Stage II and Anole. While the performance of LANTERN varies slightly depending on the model, it consistently outperforms EAGLE-2 in terms of both speed-up and mean accepted length. For instance, on the LlamaGen-XL Stage II model, LANTERN achieves a speed-up of $1.64\times$ and a mean accepted length of $2.24$, significantly higher than EAGLE-2's $0.96\times$ speed-up and $1.22$ mean accepted length. Similarly, for Anole, LANTERN achieves a speed-up of $1.17\times$ with a mean accepted length of $1.83$, compared to EAGLE-2's $0.73\times$ speed-up and $1.10$ mean accepted length.

In terms of image quality, LANTERN demonstrates minimal degradation despite the improved acceleration. For the LlamaGen-XL Stage II model, the FID decreases slightly to $46.10$ compared to the baseline of $47.60$, while the CLIP score remains stable. Precision and recall metrics indicate that LANTERN maintains a balance between individual image quality (precision) and diversity (recall).

On Anole, LANTERN similarly exhibits competitive image quality metrics, with a CLIP score of 0.3186 and a moderate decrease in HPS v2.

These results highlight the versatility of LANTERN, showcasing its ability to provide significant acceleration benefits while maintaining acceptable image quality across diverse models. This reinforces LANTERN's effectiveness as a practical and efficient approach for visual AR models.

# F    DETAILED LATENCY ANALYSIS FOR LANTERN

## F.1    ANALYSIS ON THE NUMBER OF CAPTIONS

Table 7: Actual speedup results for LANTERN across different settings of $\tau$, $k$, and $\delta$, with varying numbers of captions.

| Num Captions | Actual Speed-up $(\tau = 0, k = 1000, \delta = 0.05)$ | Actual Speed-up $(\tau = 0, k = 1000, \delta = 0.2)$ | Actual Speed-up $(\tau = 1, k = 1000, \delta = 0.1)$ | Actual Speed-up $(\tau = 1, k = 1000, \delta = 0.4)$ |
|---|---|---|---|---|
| 100 | $1.56\times$ | $2.33\times$ | $1.13\times$ | $1.73\times$ |
| 1000 | $1.56\times$ | $2.26\times$ | $1.13\times$ | $1.69\times$ |
| 2000 | $1.57\times$ | $2.27\times$ | $1.13\times$ | $1.69\times$ |
| 5000 | $1.56\times$ | $2.26\times$ | $1.13\times$ | $1.69\times$ |

Table 7 provides a statistical analysis of actual speed-up measurements conducted with different numbers of captions for MS-COCO validation captions (Lin et al., 2014) on LlamaGen-XL Stage I model (Sun et al., 2024). Across all configurations of $\tau, k$ and $\delta$, the actual speed-ups remain consistent beyond 1000 captions, with negligible differences observed. For example, for $\tau = 0, k = 1000, \delta = 0.05$, the speed-up values remain between $1.56\times$ and $1.57\times$ across 1000, 2000, and 5000 captions. Similarly, other configurations also exhibit minimal variation, confirming the reliability of using 1000 captions as a sample size.

This statistical stability justifies our choice to use 1000 captions in the main experiments, as it provides an accurate and computationally efficient estimate of actual speed-up without sacrificing reliability.

## F.2    COMPONENT-LEVEL LATENCY ANALYSIS FOR LANTERN

Table 8: Component-level latency analysis for LANTERN. Each component's latency is averaged over a single draft-and-verify process, with measurements conducted on a single RTX 3090 GPU.

| Method | Target Forward | Drafter Forward | Proximity Set $A$ Calculation |
|---|---|---|---|
| LANTERN($\delta = 0.1, k = 1000$) | $3.80 \times 10^{-2}$s | $1.08 \times 10^{-2}$s | $1.57 \times 10^{-3}$s |
| LANTERN($\delta = 0.4, k = 1000$) | | | $1.19 \times 10^{-3}$s |

In this section, we conducted experiments to evaluate the contribution of each component of LANTERN to overall latency. Specifically, we identified three primary components that significantly impact latency: (1) the target model forward pass, (2) the drafter model forward pass, and (3) the computation of the proximity set $A$. Using the LlamaGen-XL Stage I model, we measured the average latency of each component during a single draft-and-verify process across 1,000 prompts from MS-COCO validation captions (Lin et al., 2014). Experiments is conducted under two settings, $k = 1000, \delta = 0.1$ and $k = 1000, \delta = 0.4$, with the results summarized in Table 8.

Among the three key contributors to latency, the proximity set computation, which is introduced by LANTERN, required more time at lower $\delta$ values. This can be attributed to the increased number of rejections at lower $\delta$, necessitating multiple proximity set computations within a single step. However, even in the $k = 1000, \delta = 0.4$ setting, the latency of proximity set computation is $24\times$ smaller than that of the target model forward passes and about $7\times$ smaller than the drafter model forward pass.

Overall, while LANTERN introduces additional computational overhead compared to traditional speculative decoding methods, the trade-off results in significantly greater speed-up. Thus, our approach demonstrates superior actual speed-up, establishing its advantage over existing speculative decoding methods.

## F.3  Latency Comparison on Probability Distance Metrics

Table 9: Computation time comparison between TVD and JSD in LANTERN. Each component's latency is averaged over a single draft-and-verify process, with measurements conducted on a single RTX 3090 GPU. The hyperparameters for LANTERN are fixed to $k = 1000$, and $\delta = 0.4$ for TVD and $\delta = 0.2$ for JSD.

| Distance Metric | Computation Time for Distance | Total Computation Time |
|:---:|:---:|:---:|
| TVD | $1.19 \times 10^{-3}$s | $4.89 \times 10^{-2}$s |
| JSD | $4.03 \times 10^{-3}$s | $4.92 \times 10^{-2}$s |

Table 9 provides a comparison of computation times between TVD and JSD, supporting our decision to use TVD as the preferred distance metric. We use LlamaGen-XL Stage I with randomly sampled 1000 captions in MS-COCO validation captions (Lin et al., 2014) for this experiment. While both metrics produce nearly identical results in terms of mean accepted length and image quality, TVD is significantly more computationally efficient. Specifically, the computation time for TVD is $1.19 \times 10^{-3}$ seconds, which is approximately three times faster than JSD's computation time of $4.03 \times 10^{-3}$ seconds.

Although the total computation time for each decoding step (including other processes) shows a smaller difference, $4.89 \times 10^{-2}$ seconds for TVD and $4.92 \times 10^{-2}$ seconds for JSD, this difference accumulates over multiple decoding steps. For large-scale applications or models generating long sequences, this efficiency advantage becomes increasingly significant. Therefore, given the negligible impact on quality metrics and the consistent computational advantage, we choose TVD over JSD as the more practical and efficient metric for our method.

Note that the hyperparameters do not affect the computation time for the distance metric itself but do impact the total computation time. Therefore, we set the values of $\delta$ for TVD and JSD to achieve the same level of mean accepted length.

## F.4  Latency Results on Intel Gaudi 2 Accelerator

Table 10: Actual speed-up, MAL (Mean Accepted Length) for each method. $\tau = 0$ refers to the greedy decoding and $\tau = 1$ refers to the sampling with temperature 1 for generation. The actual speed-up are measured on a single Intel Gaudi 2 (96GB) accelerator and NVIDIA RTX 3090.

| Method | $\tau = 0$ | | |
|---|---|---|---|
| | Acceleration (↑) | | |
| | Speed-up (Intel Gaudi 2) | Speed-up (NVIDIA RTX 3090) | MAL |
| LlamaGen-XL Stage I | $1.00\times$ | $1.00\times$ | 1.00 |
| EAGLE-2 | $0.80\times$ | $1.29\times$ | 1.60 |
| LANTERN ($\delta = 0.05$, $k = 1000$) | $1.01\times$ | $1.56\times$ | 2.02 |
| LANTERN ($\delta = 0.2$, $k = 1000$) | $1.32\times$ | $2.26\times$ | 2.89 |
| **Method** | $\tau = 1$ | | |
| | Acceleration (↑) | | |
| | Speed-up (Intel Gaudi 2) | Speed-up (NVIDIA RTX 3090) | MAL |
| LlamaGen-XL Stage I | $1.00\times$ | $1.00\times$ | 1.00 |
| EAGLE-2 | $0.55\times$ | $0.93\times$ | 1.20 |
| LANTERN ($\delta = 0.05$, $k = 1000$) | $0.74\times$ | $1.13\times$ | 1.75 |
| LANTERN ($\delta = 0.2$, $k = 1000$) | $1.02\times$ | $1.69\times$ | 2.40 |

To demonstrate the adaptability of LANTERN across different hardware platforms, we extend our evaluation beyond NVIDIA GPUs to include the Intel Gaudi 2 accelerator (96GB). Table 10 presents the actual speed-up and Mean Accepted Length (MAL) for different methods, comparing performance on both Intel Gaudi 2 and NVIDIA RTX 3090. While the primary results in the main paper focus on CUDA-based hardware, this analysis highlights that LANTERN can effectively accelerate auto-regressive generation on Gaudi 2 as well.

The observed difference in speed-up between Gaudi 2 and NVIDIA GPUs is not indicative of a general performance gap but rather reflects the unique computational characteristics of speculative decoding. Unlike standard inference and training workloads, where Gaudi 2 has demonstrated competitive performance, speculative decoding introduces irregular execution patterns, including interleaving model forwards and non-trivial memory access patterns. These factors make speculative decoding particularly sensitive to low-level implementation details, which can vary across different accelerator architectures.

Despite the absence of Gaudi-specific optimizations in our implementation, LANTERN still achieves meaningful acceleration on this hardware. While EAGLE-2 achieves only $0.80\times$ speed-up at $\tau = 0$ and $0.55\times$ at $\tau = 1$, LANTERN improves these figures to $1.32\times$ and $1.02\times$, respectively. This demonstrates that speculative decoding can be effectively deployed on Gaudi 2 and that the core principles of LANTERN remain beneficial across diverse hardware platforms.

These results suggest that Gaudi 2 is well-suited for auto-regressive generation, including techniques such as speculative decoding. While hardware-specific optimizations could further enhance performance, the existing results already indicate that speculative decoding can be efficiently executed on Gaudi 2. As speculative decoding continues to gain traction in large-scale model deployment, further refinement of its implementation across different accelerator architectures will help maximize its benefits in diverse hardware environments.

# G    SIZE OF LATENT PROXIMITY SETS ACROSS POSITIONS

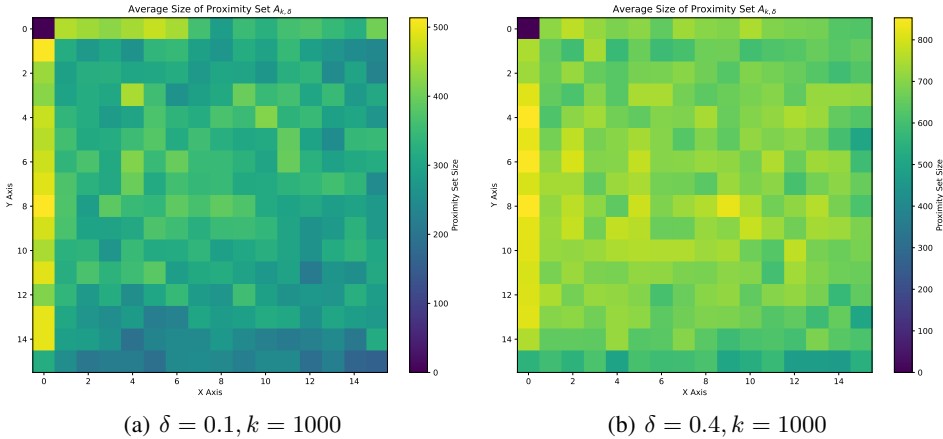

(a) $\delta = 0.1, k = 1000$             (b) $\delta = 0.4, k = 1000$

Figure 7: Average size of $A_{k,\delta}$ on different position. Note that first token is generated right after pre-fill stage, we do not conduct speculation for first token.

We conduct experiments to better understand the behavior of LANTERN by examining how the size of the latent proximity set $A$ varies depending on the position in images. For $k = 1000$ and $\delta = 0.1, 0.4$, we generate 100 images based on MS-COCO validation captions (Lin et al., 2014) with the LlamaGen-XL Stage I model for each setting. The calculated average size of $A$ on the different positions can be found in Figure 7.

As expected, larger $\delta$ values generally led to a larger size of $A$ regardless of position. Across various $\delta$ values, the size of the proximity set $A$ is consistently larger at the left edge of the image. We hypothesize that this phenomenon occurs because the left edge of the image corresponds to positions immediately following a line break, where uncertainty is higher compared to other positions. This increased uncertainty results in generally lower probabilities assigned to individual tokens, allowing a larger number of tokens to meet the threshold and be included in the set $A$.

Table 11: Precision and recall (Kynkäänniemi et al., 2019) values for varying $\delta$ and $k$ settings. The vanilla AR decoding score is also provided for reference.

| Precision / Recall | $\tau = 0$ | | | |
|---|---|---|---|---|
| | $\delta = 0.05$ | $\delta = 0.1$ | $\delta = 0.2$ | $\delta = 0.4$ |
| Vanilla AR | 0.4232 / 0.3517 | | | |
| $k = 100$ | 0.4424 / 0.3212 | 0.4510 / 0.2989 | 0.4661 / 0.2754 | 0.4689 / 0.2813 |
| $k = 300$ | 0.4488 / 0.3195 | 0.4619 / 0.2960 | 0.4696 / 0.2802 | 0.4750 / 0.2753 |
| $k = 1000$ | 0.4484 / 0.3158 | 0.4659 / 0.2939 | 0.4771 / 0.2773 | 0.4854 / 0.2682 |
| Precision / Recall | $\tau = 1$ | | | |
| | $\delta = 0.05$ | $\delta = 0.1$ | $\delta = 0.2$ | $\delta = 0.4$ |
| Vanilla AR | 0.4781 / 0.5633 | | | |
| $k = 100$ | 0.4867 / 0.5389 | 0.4796 / 0.5303 | 0.4789 / 0.5140 | 0.4825 / 0.4946 |
| $k = 300$ | 0.4856 / 0.5367 | 0.4834 / 0.5231 | 0.4894 / 0.4901 | 0.4895 / 0.4719 |
| $k = 1000$ | 0.4865 / 0.5334 | 0.4869 / 0.5172 | 0.4880 / 0.4888 | 0.4909 / 0.4497 |

Table 12: HPS v2 (Wu et al., 2023) score for different $\delta$ and $k$ settings. The baseline score for Vanilla AR decoding is provided for reference.

| HPS v2 | $\tau = 0$ | | | |
|---|---|---|---|---|
| | $\delta = 0.05$ | $\delta = 0.1$ | $\delta = 0.2$ | $\delta = 0.4$ |
| Vanilla AR | 23.18 | | | |
| $k = 100$ | 22.73 | 22.41 | 22.13 | 21.96 |
| $k = 300$ | 22.66 | 22.25 | 21.92 | 21.69 |
| $k = 1000$ | 22.62 | 22.14 | 21.69 | 21.39 |
| HPS v2 | $\tau = 1$ | | | |
| | $\delta = 0.05$ | $\delta = 0.1$ | $\delta = 0.2$ | $\delta = 0.4$ |
| Vanilla AR | 24.11 | | | |
| $k = 100$ | 24.01 | 23.94 | 23.86 | 23.75 |
| $k = 300$ | 23.97 | 23.85 | 23.70 | 23.55 |
| $k = 1000$ | 23.91 | 23.75 | 23.47 | 23.22 |

# H  TRADE-OFFS WITH ADDITIONAL METRICS

To gain a deeper understanding of the trade-offs in LANTERN, we examined precision & recall (Kynkäänniemi et al., 2019), and HPS v2 (Wu et al., 2023) in addition to FID (Heusel et al., 2017), and the results are summarized in Table 11 and 12. The experimental settings are identical to the main experiments in Table 2 and Figure 5.

The result in Table 11 shows that LANTERN achieves comparable or slightly improved precision relative to the baseline across various settings, reflecting its ability to maintain high-quality token generation. While recall decreases marginally with increasing $\delta$, this is an expected trade-off due to the relaxed acceptance condition, which prioritizes sampling speed. The precision/recall results demonstrate that our method strikes a reasonable balance between quality and diversity across different hyperparameter configurations.

To further quantify the aesthetic quality of generated images, we evaluate our approach using HPS v2. Table 12 shows that while there is a slight reduction in aesthetic quality compared to the baseline, the trade-off is well-justified by the significant improvements in generation speed. This aligns with the intended design of LANTERN, which emphasizes efficiency while preserving acceptable quality.

## I    TEXT PROMPTS FOR QUALITATIVE SAMPLES

The prompts listed below were used to generate Figure 4, with the images in the figure generated sequentially from left to right using Prompt 1 through Prompt 9.

---

**Prompt 1**

A serene lake reflecting the colors of a sunset sky with distant mountains and a few birds flying across the sky.

---

**Prompt 2**

A cozy bedroom with soft, warm lighting, a bed with fluffy pillows, a small bookshelf filled with books, and a potted plant on the windowsill.

---

**Prompt 3**

A beautiful stained glass window with sunlight streaming through in a church, casting colorful patterns on the stone floor and pews.

---

**Prompt 4**

A close-up of a single pink rose in full bloom, with soft, layered petals and gentle sunlight illuminating its delicate curves.

---

**Prompt 5**

A classic still life of a bowl of fresh fruit, including apples, oranges, and grapes, with light softly highlighting the textures of each fruit.

---

**Prompt 6**

A high-fashion portrait of a woman with vibrant, artistic makeup and bold accessories, wearing a modern, avant-garde outfit against a plain studio background.

---

**Prompt 7**

A close-up of a man gazing thoughtfully out of a rain-covered window, with soft reflections and water droplets creating a melancholic, introspective atmosphere.

---

**Prompt 8**

A close-up of a small bird perched on a snow-covered branch, with soft, fluffy feathers and a delicate beak, illuminated by gentle winter sunlight.

---

**Prompt 9**

A close-up of a wolf with intense, focused eyes and thick gray fur, staring directly at the camera, set against a blurred forest background.

---

# J QUALITATIVE RESULTS

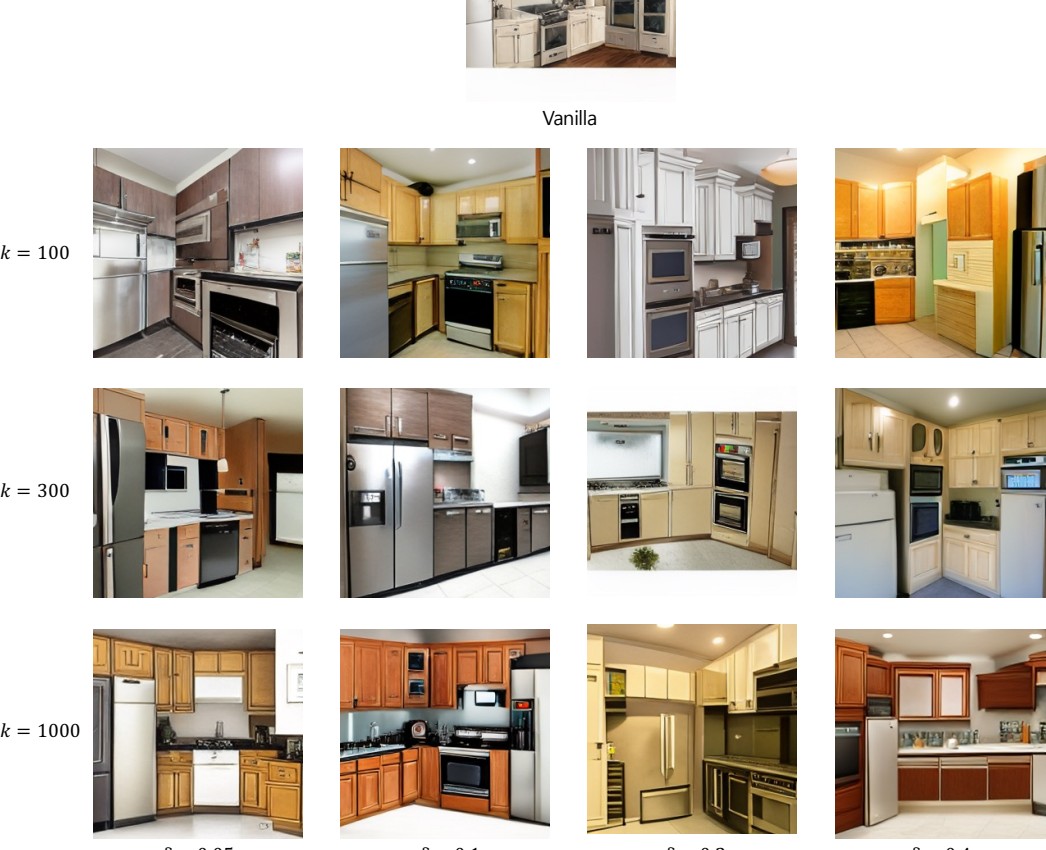

Figure 8: Qualitative sample for the changes in the generated images according to various $\delta \in \{0.05, 0.1, 0.2, 0.4\}$ and $k \in \{100, 300, 1000\}$ at $\tau = 1$. Input prompt is 'A kitchen with a refrigerator, stove and oven with cabinets'. Target model is LlamaGen-XL Stage I.

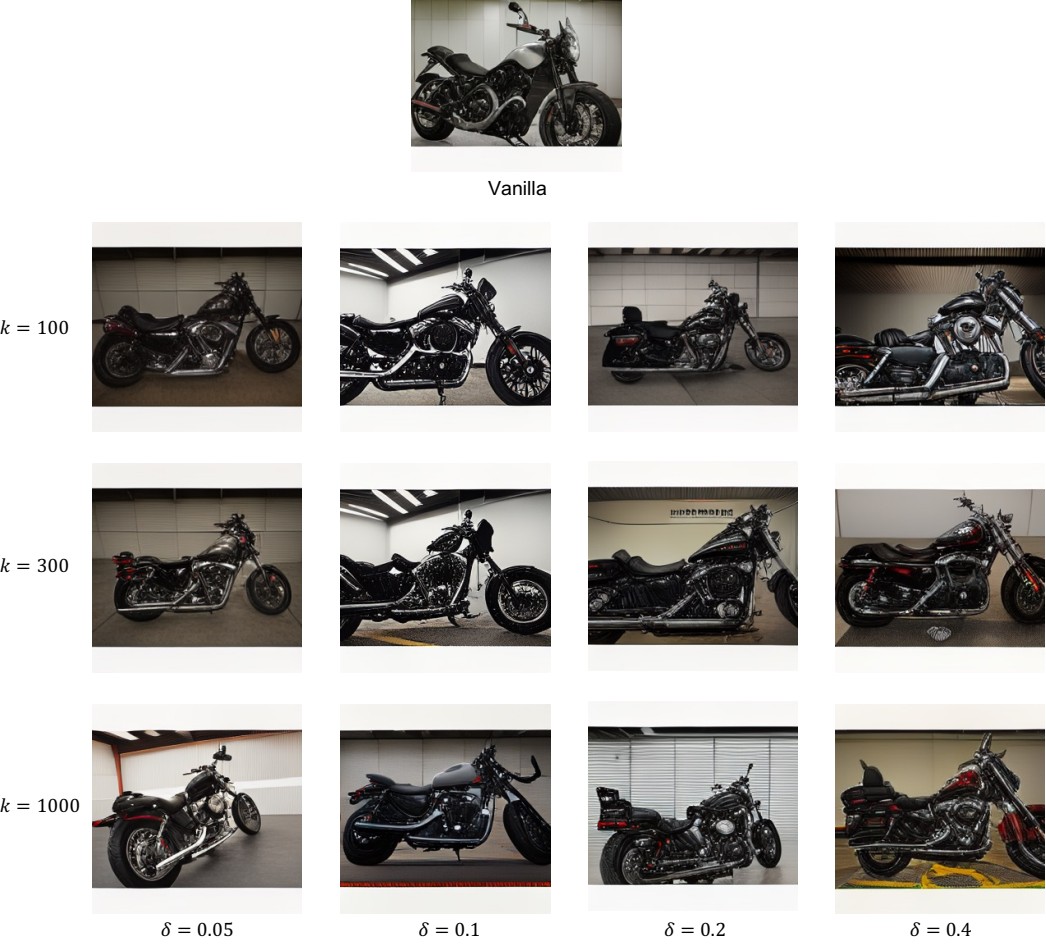

Figure 9: Qualitative sample for the changes in the generated images according to various $\delta \in \{0.05, 0.1, 0.2, 0.4\}$ and $k \in \{100, 300, 1000\}$ at $\tau = 0$. The input prompt is 'A motorcycle parked in a parking space next to another motorcycle.'. The target model is LlamaGen-XL Stage I.

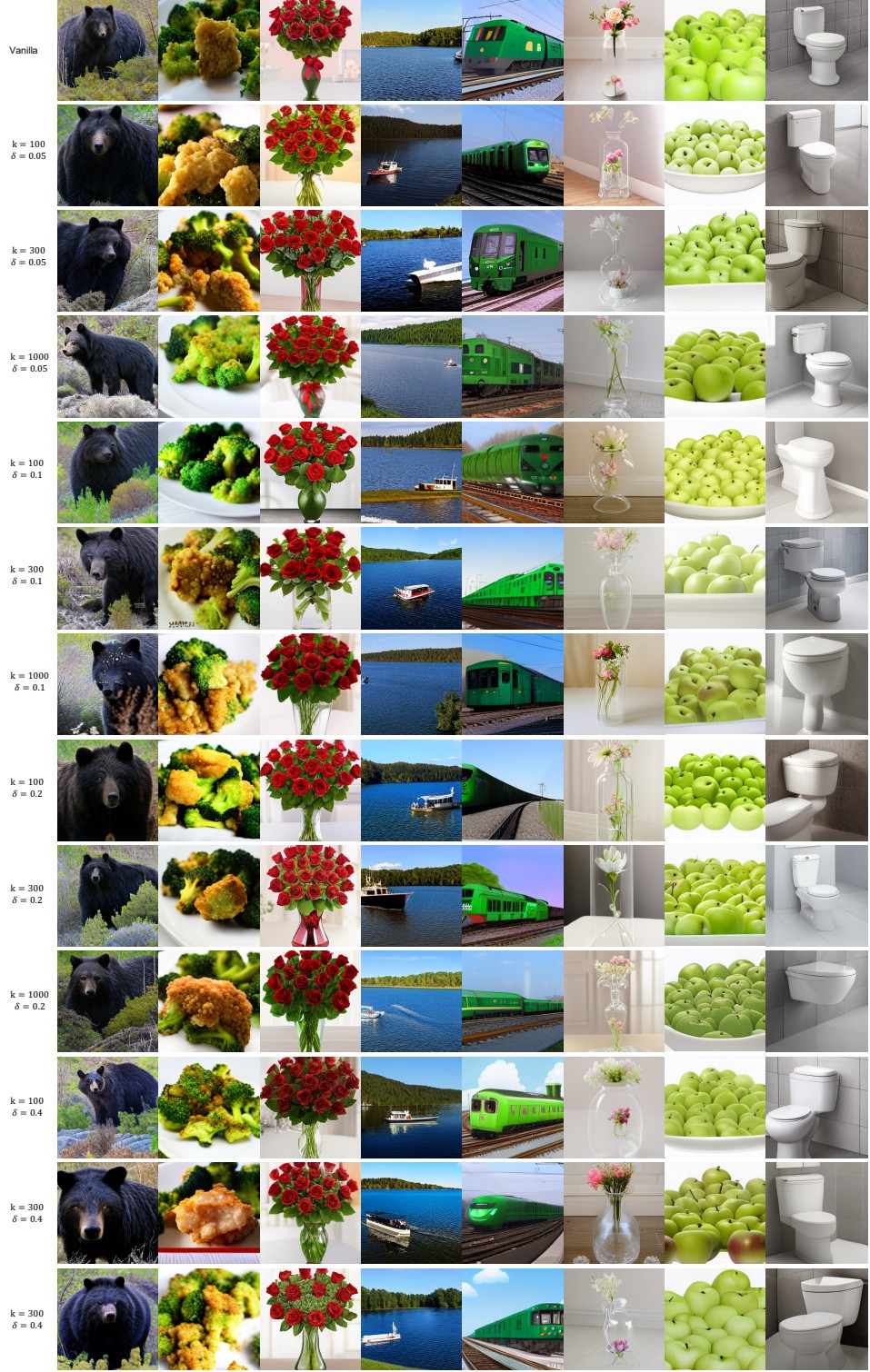

Figure 10: Additional qualitative samples with various $k$ and $\delta$ at $\tau = 1$. The target model is LlamaGen-XL Stage I, and MS-COCO validation captions are used. Images within the same column are generated using the same text prompt.

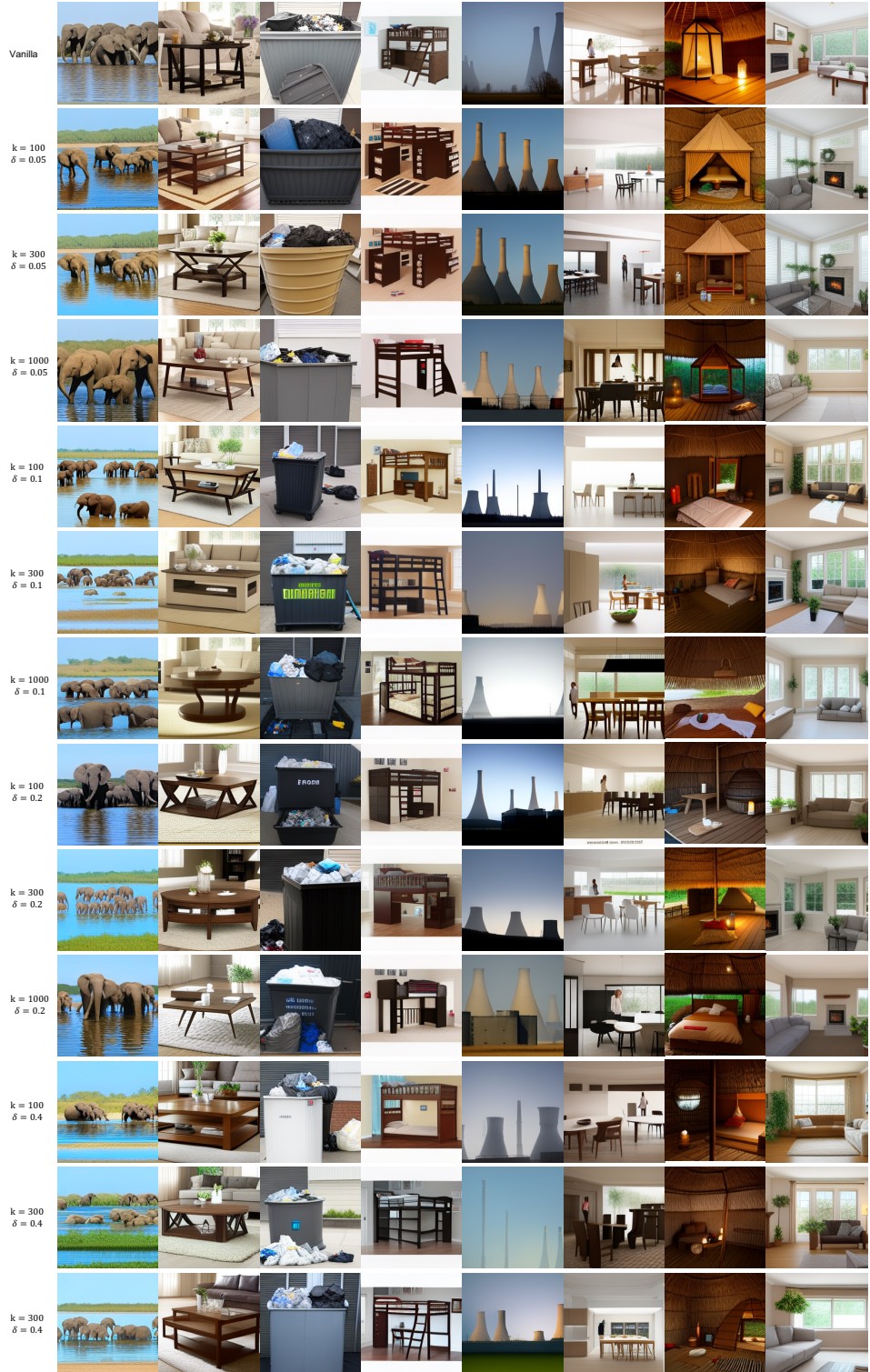

Figure 11: Additional qualitative samples with various $k$ and $\delta$ at $\tau = 0$. The target model is LlamaGen-XL Stage I, and MS-COCO validation captions are used. Images within the same column are generated using the same text prompt.

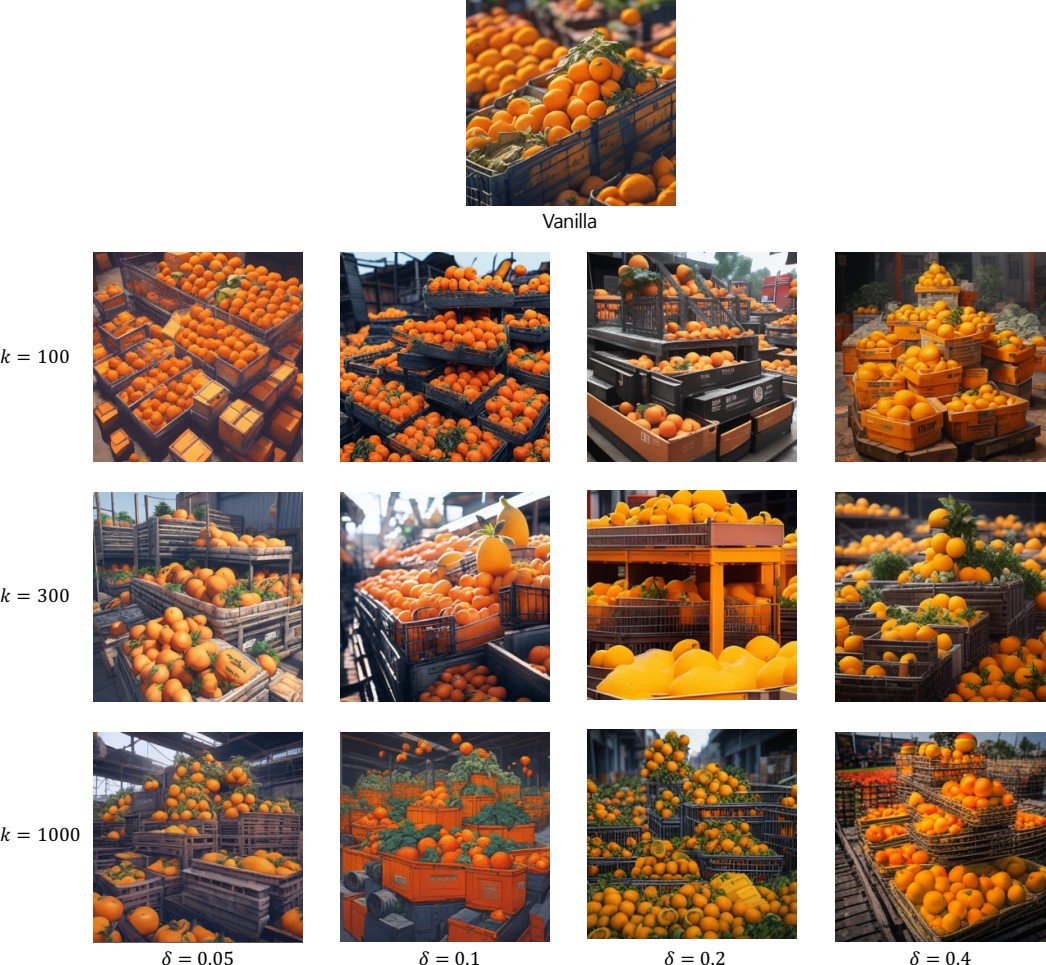

Figure 12: Qualitative sample for the changes in the generated images according to various $\delta \in \{0.05, 0.1, 0.2, 0.4\}$ and $k \in \{100, 300, 1000\}$ at $\tau = 1$. The input prompt is 'A pile of oranges in crates topped with yellow bananas.' The target model is LlamaGen-XL Stage II.

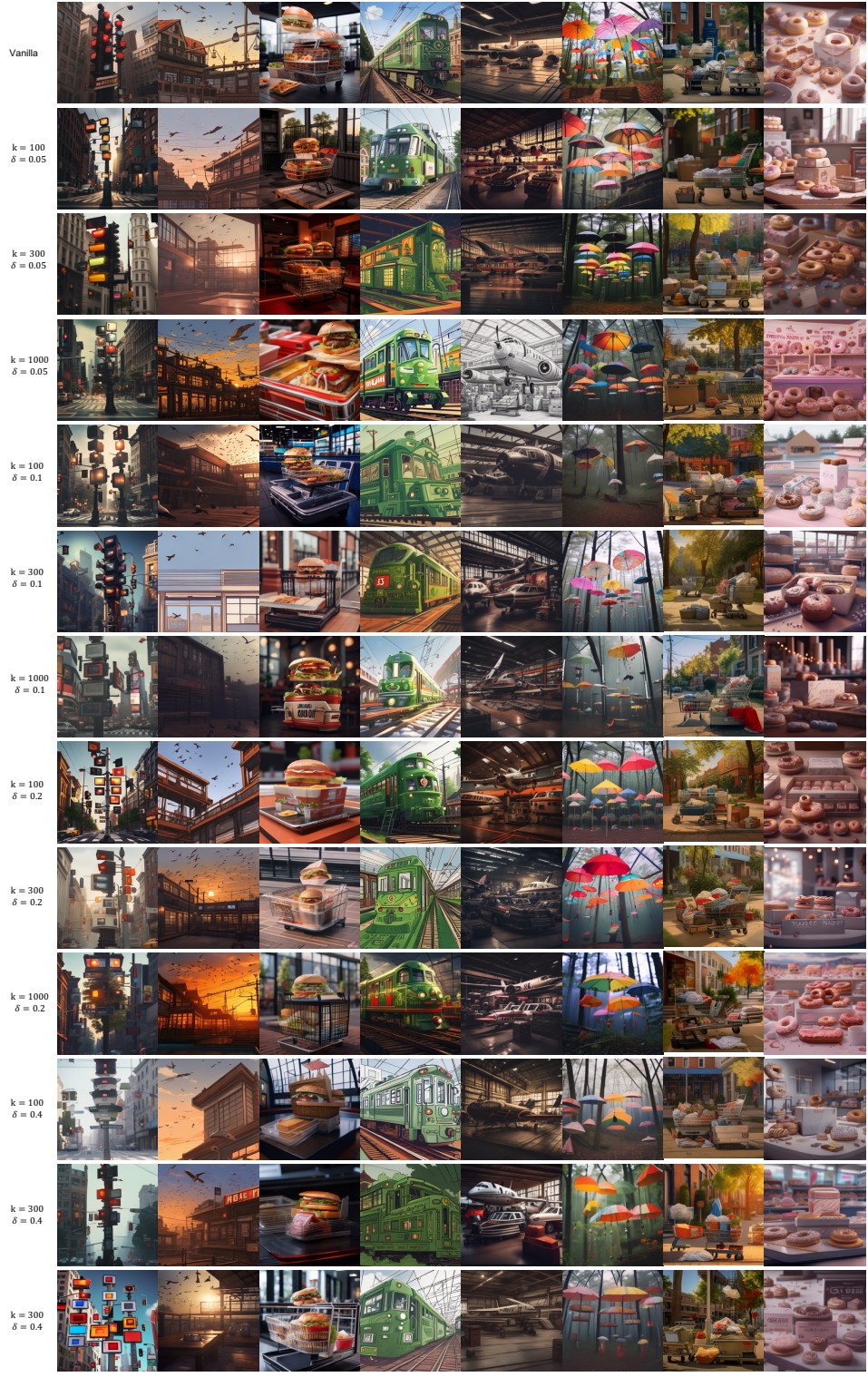

Figure 13: Additional qualitative samples with various $k$ and $\delta$ at $\tau = 1$. The target model is LlamaGen-XL Stage II, and MS-COCO validation captions are used. Images within the same column are generated using the same text prompt.

