# OpenReview forum: "LANTERN: Accelerating Visual Autoregressive Models with Relaxed Speculative Decoding"
_ICLR.cc/2025/Conference — ICLR 2025 Poster_

### Official Review · Reviewer_NFR2 · 2024-10-31

**Soundness:** 3
**Presentation:** 3
**Contribution:** 3
**Rating:** 8
**Confidence:** 4

**Summary:**

This paper tackles the challenge of transferring speculative decoding from LLMs to autoregressive image generation models. The authors observe that the nature of image data causes image tokens to exhibit selection ambiguity, resulting in high rejection rates and poor acceleration effects in speculative decoding. Hence  they propose using the proximity set of image tokens as proxies to speculate on accepting a predicted image token, and introduce a combinatorial optimization strategy for selecting the set to ensure image quality remains intact.  Extensive experiments confirm significant acceleration with minimal quality loss.

**Strengths:**

The paper's structure and presentation are well-crafted. It identifies the phenomenon of "token ambiguity" and provides a viable solution. Extensive experiments, particularly the ablation studies, demonstrate the effectiveness and scalability of the method. It offers valuable insights for research on inference acceleration in AR text-to-image generation. Although the idea of using a set to proxy a single point has been extensively researched, the authors have elegantly applied it to the sampling decision process and achieved excellent results.
 Simplicity yet effectiveness is crucial.

**Weaknesses:**

1.For a specific image token, the algorithm for finding its proximity set , specifically "Find the neighborhood (i.e., Appendix B, Line 9)," lacks detailed discussion on efficiency. For example, for each quantized image token, are the corresponding sets A and B precomputed or dynamically calculated during inference? How much time does this part take? I recommend the authors supplement this section with analysis and discussion.

2.In Line 105, there might be a misstatement. EAGLE-2 is an acceleration decoding method, not a drafter.

3.To demonstrate the generalizability of the observation of "token ambiguity," i.e., that this phenomenon is present in most AR image generation models, the paper should include experiments on a broader range of AR image generation models beyond just LLaMAGen. This would enhance the experimental completeness and provide stronger evidence for the generality of the observed issue.

**Questions:**

1.There's a suggestion that the authors could provide the details of "Find the neighborhood" in Appendix A.

2.A suggestion is (since the experimental section of the current paper is already sufficient, this is just a gentle suggestion) that the authors could discuss more about the differences in token ambiguity between text generation and image generation. For example, in LLaMAGen, what common properties do tokens within the same set have? What are the patterns in the sizes of sets A and B corresponding to different image tokens? These discussions could help in designing further algorithms.

3.Could validation experiments be conducted on other AR models? This is also a optional suggestion aimed at enhancing the generalization and robustness of the method.

**Details Of Ethics Concerns:**

There are no ethical concerns in this paper.

---

> ### Author Response · Authors · 2024-11-21
> **Response to Reviewer NFR2 (1)**
>
> First of all, we sincerely appreciate the detailed review of our paper and the constructive feedback provided. Thank you for highlighting the clarity of our presentation and recognizing the novelty and elegance of our approach to tackle token selection ambiguity. We believe your feedback has significantly strengthened our work. To address the concerns and questions, we have conducted additional experiments and revised the manuscript as follows:
>
> ---
> ### **Weakness 1 & Question 1: Finding Neighborhood**
>
> To identify the proximity sets A and B of quantized image tokens, we employed both a dynamic approach and precomputation. Specifically:
>
> - **Precomputation of $B$:** The set $B$ comprises the $k$ nearest tokens in the latent space with $\ell_2$ distance for each quantized image token. Since $B$ is independent of token probabilities (and preceding tokens), it is precomputed and stored as tensors before inference begins.
> - **Dynamic Calculation of $A$:** The set $A$ is a subset of $B$, determined dynamically based on token probabilities and an upper bound $\delta$. This ensures $A$ adapts to the current decoding context.
>
> We have summarized the overall procedure as pseudocode and included it as Appendix D.2 in the revised manuscript.
>
> To evaluate the impact of these computations on overall latency, we measured the time required to compute $A$ during a single inference step and compared it with the forward times for the target model and drafter. Target model is LlamaGen Stage I and randomly selected 1000 caption from MS-COCO 2014 validation set are used. The results are summarized below:
>
> |  | Target forward | Drafter forward | Proximity set $A$ calculation |
> | --- | --- | --- | --- |
> | LANTERN $(\delta=0.40, k=1000)$ | $3.80\times10^{-2}$s | $1.08\times10^{-2}$s | $1.19\times10^{-3}$s |
>
> Our analysis shows that the time required to compute $A$ is 32 times smaller than drafter model forward and 9.1 times smaller than target model forward for $k=1000, \delta=0.4$, which is negligible in entire process. Furthermore, the significant speedup enabled by $A$ also confirms the efficiency of our method.
>
> These findings have been added to Appendix F.2, along with further discussion on implementation details. We appreciate your suggestion, which has enhanced the clarity and comprehensiveness of our work.
>
> ---
>
> ### **Weakness 2: Misstatement in Line 105**
>
> We acknowledge the incorrect description of EAGLE-2 as a drafter in Line 105. This has been corrected to EAGLE-2 as a base speculative decoding method at Line 105 in the revised manuscript. Thank you for pointing this out, allowing us to rectify this misstatement.
>
> ### **Weakness 3 & Question 3: Generalizability of the Token Selection Ambiguity and LANTERN**
>
> Thank you for your valuable suggestion on the generalizability of token selection ambiguity and our method LANTERN to other visual AR models. To demonstrate the generalizability of the token selection ambiguity and our method, we conducted additional experiments on other visual AR models, including LlamaGen Stage II model and Anole [1] with MS-COCO 2017 Validation set.
>
> | Models | Average Top-1 probabilities | Average Top-10 probabilities | Drafter Test Accuracy |
> | --- | --- | --- | --- |
> | Vicuna-13B | $0.787$ | $0.989$ | $84.66\%$ |
> | LlamaGen-XL | $0.206$ | $0.520$ | $38.80\%$ |
> | Anole | $0.064$ | $0.204$ | $27.60\%$
>  |
>
> As shown in the table, like LlamaGen, Anole exhibits low average top-1 and top-10 probabilities, with values of $0.064$ and $0.204$, respectively. However, these probabilities are even lower than those of LlamaGen ($0.206$ and $0.520$), suggesting that Anole faces a more severe degree of token selection ambiguity. This increased ambiguity also affects drafter performance, with Anole’s drafter achieving a test accuracy of only $27.60\%$, compared to $38.80\%$ for LlamaGen.
>
> By extending the analysis to Anole, we confirm that the token selection ambiguity is a recurring challenge across different models, with varying degrees of severity. This further substantiates our claim that token selection ambiguity is a critical factor influencing the performance of speculative decoding. The results have been incorporated into the Figure 2(b) and (c) in the revised manuscript.

---

> > ### Author Response · Authors · 2024-11-21
> > **Response to Reviewer NFR2 (2)**
> >
> > | LlamaGen Stage II, $\tau=1$ | Speedup | Mean Accepted Length | FID | CLIPScore | Precision / Recall | HPS v2 |
> > | --- | --- | --- | --- | --- | --- | --- |
> > | Vanilla AR | $1.00\times$ | $1.00\times$ | $47.60$ | $0.2939$ | $0.4138$  / $0.5648$ | $23.84$ |
> > | EAGLE-2 | $0.96\times$ | $1.22\times$ | - | - | - | - |
> > | LANTERN ($\delta=0.40, k=1000)$ | $1.64\times$ | $2.24\times$ | $46.10$ | $0.2925$ | $0.4704$  / $0.5222$ | $23.06$ |
> >
> > | Anole, $\tau=1$ | Speedup | Mean Accepted Length | FID | CLIPScore | Precision / Recall | HPS v2 |
> > | --- | --- | --- | --- | --- | --- | --- |
> > | Vanilla AR | $1.00\times$ | $1.00\times$ | $20.27$ | $0.3215$ | $0.6552$  / $0.6398$ | $23.52$ |
> > | EAGLE-2 | $0.73\times$ | $1.10\times$ | - | - | - | - |
> > | LANTERN ($\delta=0.50, k=100)$ | $1.17\times$ | $1.83\times$ | $23.40$ | $0.3186$ | $0.6026$  / $0.6178$ | $22.92$ |
> >
> > The results show that LANTERN consistently achieves substantial acceleration (with speed-ups of $1.71\times$ and $1.60\times$ over EAGLE-2 on LlamaGen Stage II and Anole, respectively) while maintaining competitive image quality. However, its performance, in both acceleration and image quality, varies depending on the degree of token selection ambiguity inherent to the underlying model. In particular, Anole's results are somewhat less favorable compared to those of LlamaGen, a difference attributable to the distinct characteristics of the two models.
> >
> > As discussed earlier, Anole exhibits a higher degree of token selection ambiguity compared to LlamaGen, resulting in lower drafter performance. This discrepancy impacts the outcomes for both EAGLE-2 and LANTERN on Anole. Nevertheless, it is essential to note that this variation does not represent a limitation of LANTERN itself but rather reflects the impact of model-specific factors, such as token selection ambiguity. Despite these challenges, LANTERN continues to outperform EAGLE-2 on Anole in terms of acceleration and maintains reasonable image quality, highlighting its robustness across different models.
> >
> > Furthermore, these findings open avenues for future research. Improving drafter performance through enhanced architectures or training approaches that address token selection ambiguity could further optimize results for models with higher levels of ambiguity. While the current results affirm LANTERN’s robustness and versatility, addressing these factors could lead to even greater performance improvements in the future. We sincerely thank you for raising these important questions, which have allowed us to explore these new research directions and better understand the role of token selection ambiguity in model performance.
> >
> > We have included the these results to appendix E in the revised manuscript, as we aimed to minimize significant changes to the main content of the manuscript. However, if you feel that incorporating these results into the main table would enhance clarity and comprehensiveness, we would be happy to expand the main table in the next revision. We sincerely appreciate all of your valuable feedback and remain committed to improving the manuscript in line with your suggestions.

---

> > > ### Author Response · Authors · 2024-11-21
> > > **Response to Reviewer NFR2 (3)**
> > >
> > > ### **Question 2: Further Analysis on Token Ambiguity**
> > >
> > > We sincerely thank you for your thoughtful suggestion and for recognizing the potential value of further discussions on token ambiguity. We conducted additional analysis to better understand the nature of token ambiguity:
> > >
> > > 1. **Visualization of Proximity Tokens for Visual AR Models:** We visualized how tokens in $B$ decode into image patches with LlamaGen Stage II model. The visualizations reveal that tokens in $B$ correspond to visually similar patches, validating the hypothesis of latent proximity in image generation. The results have been incorporated into Appendix C.2 and the same figures can be found in this anonymous links: [Sample 1](https://postimg.cc/K34P8YWq) [Sample 2](https://postimg.cc/YGP48zXR).
> > > 2. **Proximity Token Sets in Language AR Models:** We also conducted an investigation into proximity token sets for language models (Vicuna-7B). Since language models do not utilize encoders like VQVAE in visual AR models, we examined the proximity of tokens based on their input embeddings (representations obtained through the embedding layer). As expected, we observed that, unlike visual AR models, tokens in language models do not display clear semantic similarity among those considered "close" in the embedding space. While this finding may not be surprising, it underscores a fundamental difference in how token relationships are structured across the two domains.
> > >
> > >
> > >     | Token: hi | Top-1 | Top-2 | Top-3 | Top-4 | Top-5 |
> > >     | --- | --- | --- | --- | --- | --- |
> > >     | Nearest | \_Portály | \_Mediabestanden | <0x4C> | <0x6B> | <0x49> |
> > >     | Farthest | [Line Seperator] | [Object Replacement Character] | \_infinitely | \_firewall | \_sooner |
> > >
> > >     | Token: act | Top-1 | Top-2 | Top-3 | Top-4 | Top-5 |
> > >     | --- | --- | --- | --- | --- | --- |
> > >     | Nearest | \_Mediabestanden | oreferrer | <0x24> | <0x71> | <0x54> |
> > >     | Farthest | [Object Replacement Character] | \_Bruno | \_Ernst | \_Santos | \_firewall |
> > > 3. **Patterns in $A$ Sizes:** Since $A$ is determined dynamically, we analyzed how its size varies across different positions in the generated image. With LlamaGen Stage I model and randomly sampled 100 captions in MS-COCO 2014 validation captions, we calculate avrage size of $A$ with respect to its position in image. First, at high $\delta$, the size of $A$ is generally large. Additionally, it was observed that the average size of $A$ tend to be larger at the left end of image. We hypotheize that this is due to the higher uncertainty at the left side caused by line change of image. Since the probabilities assigned to individual tokens are relatively smaller at the left end, allowing more tokens to be included for the same $\delta$, which increases the average size of $A$. You can find actual heatmap for $k=1000$ and $\delta=0.1, 0.4$ in following anonymous links: [delta=0.1](https://postimg.cc/94rw9kb0) [delta=0.4](https://postimg.cc/LhzYjLtt).
> > >
> > > These insights have been incorporated into the revised manuscript, as Appendix C and G.
> > >
> > > We hope these revisions and additional experiments address your concerns comprehensively. We deeply value your thoughtful and constructive input, which has played a pivotal role in enhancing the rigor and quality of our work. It has been a privilege to engage with your insightful questions and critiques. Thank you once again for your careful consideration of our manuscript and for giving us the opportunity to further refine our contribution.
> > >
> > > **References**
> > >
> > > [1] Chern, Ethan, et al. "Anole: An open, autoregressive, native large multimodal models for interleaved image-text generation." *arXiv preprint arXiv:2407.06135* (2024).

---

> ### Comment · Reviewer_NFR2 · 2024-11-23
> **Response**
>
> Thank you for the detailed feedback and experiments. I think most of my concerns have been addressed, so I will raise my score to an Accept.

---

> > ### Author Response · Authors · 2024-11-25
> >
> > Thank you for taking the time to provide such thoughtful feedback and for your consideration of our work. We’re glad to hear that your concerns have been addressed, and we truly appreciate your support and the raised score.

---

### Official Review · Reviewer_Axtn · 2024-11-01

**Soundness:** 3
**Presentation:** 3
**Contribution:** 2
**Rating:** 6
**Confidence:** 3

**Summary:**

This paper proposed speculative decoding for AR Image generation, AR models tend to assign uniformly low probabilities across a wide range of tokens, making it difficult to select the most appropriate token during decoding.

To solve this problem, the author introduce LANETERN (Latent Neighbor Token Acceptance Relaxation), leveraging the interchangeability of the tokens and relaxing the acceptance for decoding.

The benefits of LANETREN can be found at generating speed with comparable quality.

**Strengths:**

1. The motivation behind the author’s proposed approach is straightforward: in standard sampling processes, an excessively high token rejection rate can significantly slow down generation. The author suggests a more relaxed policy to mitigate this issue, thereby improving generation speed.

2. To mitigate the distributional shift introduced by the lenient sampling approach, the author proposes a constraint rule for sampling based on total variation.

3. Through extensive ablation experiments, the author demonstrates how the lenient sampling scheme influences sampling speed and elucidates the specific roles of various hyperparameters within this scheme.

**Weaknesses:**

My main concern lies in whether the metrics employed in the experimental section sufficiently capture the effectiveness of the proposed approach, particularly in Table 3 and Section 4.2.3. For evaluating basic generation quality, would it be possible to provide precision and recall (P/R) instead of FID? This change could allow for a clearer view of how LANTERN and limited distribution divergence affect and restore distribution. Additionally, since LlamaGen is used as the base model, could the authors include some modern image quality scores, such as PickScore or HPS, to better quantify aesthetic quality loss?

**Questions:**

see weakness.

In summary, the author presents a promising sampling scheme for AR models, which improves sampling speed at the cost of some image quality.

The primary concern is that the author’s experiments lack sufficient evidence to demonstrate that the loss in image quality is within an acceptable range. Given this limitation in quality, I believe the impact of the work may be constrained. Therefore, I am inclined to recommend rejection.

---

> ### Author Response · Authors · 2024-11-21
> **Response to Reviewer Axtn (1)**
>
> We sincerely thank you for your thoughtful feedback and constructive suggestions. We appreciate your recognition of the motivation and soundness of our approach, as well as your valuable insights on evaluation metrics and image quality. Your comments provide important guidance, and we address them in detail below.
>
> ---
> ### **Metrics for Evaluating Basic Generation Quality (Precision and Recall)**
>
> We appreciate your suggestion to include precision and recall (P/R) metrics to better understand the quality and diversity trade-offs in our approach. To this end, we have evaluated **LANTERN** with varying values of the hyperparameters $\delta$ and $k$, and compared its performance to the baseline. The results are as follows:
>
> - **Precision / Recall for standard AR**: $0.4781$ / $0.5633$
> - **LANTERN ($\tau=1.0$)**
>
>
>     | Precision / Recall | $\delta=0.05$ | $\delta=0.1$ | $\delta=0.2$ | $\delta=0.4$ |
>     | --- | --- | --- | --- | --- |
>     | $k=100$ | $0.4867$  / $0.5389$ | $0.4796$  / $0.5303$ | $0.4789$  / $0.5140$ | $0.4825$  / $0.4946$ |
>     | $k=300$ | $0.4856$  / $0.5367$ | $0.4834$  / $0.5231$ | $0.4894$  / $0.4901$ | $0.4895$  / $0.4719$ |
>     | $k=1000$ | $0.4865$  / $0.5334$ | $0.4869$  / $0.5172$ | $0.4880$  / $0.4888$ | $0.4909$  / $0.4497$ |
>
> The results demonstrate that **LANTERN** achieves comparable or slightly improved precision relative to the baseline across various settings, highlighting its ability to maintain the quality of generated images. While recall decreases slightly with increasing $\delta$, this indicates that the quality of individual images is preserved, though there is a modest reduction in diversity.
>
> The slight decrease in recall can be attributed to token selection ambiguity in the drafter. When the drafter is not optimally trained, it may struggle to produce sufficiently diverse or accurate predictions. Consequently, increasing the acceptance probability enhances acceleration but can lead to reduced image diversity. Nevertheless, this trade-off does not significantly impact overall image quality, as evidenced by consistent FID, Precision, and HPS v2 scores (in the next section). These precision and recall results illustrate that our method effectively balances quality and diversity across different hyperparameter configurations.
>
> Overall, these evaluations provide further evidence that **LANTERN** maintains generation quality within acceptable bounds while achieving substantial speed improvements.
>
> ---
> ### **Incorporation of Modern Image Quality Metrics (PickScore and HPS v2)**
>
> To further quantify the aesthetic quality of generated images, we have evaluated our approach using **HPS v2** [1], as suggested. Evaluating **PickScore** [2] was challenging due to its high time requirements, which involve measuring Elo ratings or win rates for each individual samples to properly evaluate it. Both PickScore and HPSv2 are metrics for assessing human preference; however, HPSv2 reportedly aligns more closely with human preference scores. Therefore, we chose to use only HPSv2 in our evaluation. This modern image quality metrics provide a more holistic measure of perceptual quality and aesthetic appeal. The updated findings are as follows:
>
> | HPS v2
> Vanilla AR : 24.11 | $\delta=0.05$ | $\delta=0.1$ | $\delta=0.2$ | $\delta=0.4$ |
> | --- | --- | --- | --- | --- |
> | $k=100$ | $24.01$ | $23.94$ | $23.86$ | $23.75$ |
> | $k=300$ | $23.97$ | $23.85$ | $23.70$ | $23.55$ |
> | $k=1000$ | $23.91$ | $23.75$ | $23.47$ | $23.22$ |
>
> These evaluations confirm that while there is a slight reduction in aesthetic quality compared to the baseline, the trade-off is well-justified by the significant improvements in generation speed. This aligns with the intended design of LANTERN, which emphasizes efficiency while preserving acceptable quality.
>
> The results have been incorporated to our main table (Table 2 in the revised manuscript) and full results have been appended to Appendix H. The added P/R analysis complements our FID results, offering a more comprehensive view of the method’s performance.

---

> > ### Author Response · Authors · 2024-11-21
> > **Response to Reviewer Axtn (2)**
> >
> > ### **Impact of Quality Loss and Robustness Across Diverse Model**
> >
> > To ensure that the observed quality loss remains within an acceptable range, we conducted additional experiments across diverse AR model including LlamaGen stage II and Anole [3] with MS-COCO 2017 Validation set. The results validate the robustness of our method across different settings and highlight that the trade-off between speed and quality is consistent and favorable. These are the tables of additional experiment results :
> >
> > | LlamaGen Stage II, $\tau=1$ | Speedup | Mean Accepted Length | FID | CLIPScore | Precision / Recall | HPS v2 |
> > | --- | --- | --- | --- | --- | --- | --- |
> > | Vanilla AR | $1.00\times$ | $1.00\times$ | $47.60$ | $0.2939$ | $0.4138$  / $0.5648$ | $23.84$ |
> > | EAGLE-2 | $0.96\times$ | $1.22\times$ | - | - | - | - |
> > | LANTERN ($\delta=0.40, k=1000)$ | $1.64\times$ | $2.24\times$ | $46.10$ | $0.2925$ | $0.4704$  / $0.5222$ | $23.06$ |
> >
> > | Anole, $\tau=1$ | Speedup | Mean Accepted Length | FID | CLIPScore | Precision / Recall | HPS v2 |
> > | --- | --- | --- | --- | --- | --- | --- |
> > | Vanilla AR | $1.00\times$ | $1.00\times$ | $20.27$ | $0.3215$ | $0.6552$  / $0.6398$ | $23.52$ |
> > | EAGLE-2 | $0.73\times$ | $1.10\times$ | - | - | - | - |
> > | LANTERN ($\delta=0.50, k=100)$ | $1.17\times$ | $1.83\times$ | $23.40$ | $0.3186$ | $0.6026$  / $0.6178$ | $22.92$ |
> >
> > The results demonstrate that LANTERN consistently delivers significant acceleration (achieving $1.71\times$ and $1.60\times$ speed-ups compared to EAGLE-2 on LlamaGen Stage II and Anole, respectively) while maintaining competitive image quality. However, its performance, both in acceleration and image quality, can vary based on the degree of token selection ambiguity inherent to the model. In particular, the results on Anole are slightly less favorable compared to LlamaGen, which can be explained by the distinct characteristics of these models.
> >
> > Anole exhibits much lower next-token prediction probabilities (average top-$1$ probability of $0.064$ and average top-$10$ probability of $0.204$) compared to LlamaGen ($0.206$ and $0.520$, respectively), as shown in Figure 2(c) of the revised manuscript. This highlights that Anole faces a more severe degree of token selection ambiguity, which directly affects the drafter’s ability to provide accurate predictions. Consequently, the drafter trained for Anole achieves a test accuracy of only $27.60\%$, significantly lower than the $38.80\%$ test accuracy achieved by the drafter for LlamaGen, as presented in Figure 2(b). This gap in drafter performance impacts the results for both EAGLE-2 and LANTERN on Anole.
> >
> > It is important to clarify that this variation is not a limitation of LANTERN itself but rather a reflection of model-specific factors, such as token selection ambiguity. Even under these challenging conditions, LANTERN still outperforms EAGLE-2 on Anole in terms of acceleration and maintains acceptable image quality, demonstrating its robustness across different models.
> >
> > These findings also suggest opportunities for further research. Enhancing drafter performance through better architectural designs or incorporating training methods that account for token selection ambiguity could improve results across models with significant ambiguity issues. While the current results validate LANTERN’s robustness and broad applicability, addressing these aspects could unlock even greater performance improvements in the future.
> >
> > We have included the these results to appendix E in the revised manuscript, as we aimed to minimize significant changes to the main content of the manuscript. However, if you feel that incorporating these results into the main table would enhance clarity and comprehensiveness, we would be happy to expand the main table in the next revision. We sincerely appreciate all of your valuable feedback and remain committed to improving the manuscript in line with your suggestions.
> >
> > Once again, we would like to express our heartfelt gratitude for your detailed review and constructive suggestions. Your feedback has not only helped us identify areas for improvement but has also motivated us to refine our work further. We hope that the additional experiments, analyses, and revisions we have outlined above adequately address your concerns. Please do not hesitate to let us know if there are any other aspects we should clarify or improve. We sincerely value your time and effort in reviewing our submission and are deeply appreciative of the insights you have provided.

---

> > > ### Author Response · Authors · 2024-11-21
> > > **Response to Reviewer Axtn**
> > >
> > > **References**
> > >
> > > [1] Wu, Xiaoshi, et al. "Human preference score v2: A solid benchmark for evaluating human preferences of text-to-image synthesis." *arXiv preprint arXiv:2306.09341* (2023).
> > >
> > > [2] Kirstain, Yuval, et al. "Pick-a-pic: An open dataset of user preferences for text-to-image generation." *Advances in Neural Information Processing Systems* 36 (2023): 36652-36663.
> > >
> > > [3] Chern, Ethan, et al. "Anole: An open, autoregressive, native large multimodal models for interleaved image-text generation." *arXiv preprint arXiv:2407.06135* (2024).

---

### Official Review · Reviewer_56i9 · 2024-11-01

**Soundness:** 3
**Presentation:** 3
**Contribution:** 3
**Rating:** 6
**Confidence:** 3

**Summary:**

The work presents a novel approach to enhance the efficiency of image generation using Auto-Regressive (AR) models, which traditionally suffer from slow sequential processing. The authors introduce LANTERN, a method that leverages speculative decoding with a relaxed acceptance condition to significantly speed up inference while maintaining image quality. By utilizing a smaller drafter model to predict multiple tokens simultaneously, LANTERN addresses the challenges of token selection ambiguity inherent in visual AR models. The results demonstrate notable improvements in speed and efficiency compared to baseline methods, highlighting the potential of LANTERN to advance the capabilities of AR models in generating high-quality images.

**Strengths:**

The methodology of this paper is clear and comprehensible, and the research question it addresses is interesting. Although the overall approach is relatively straightforward, I believe the findings are still valuable.

**Weaknesses:**

1. My primary concern is that the paper lacks an evaluation of the image generation quality. The authors present several generated images, but these images are stylistically very similar and exhibit poor consistency with the accompanying generated text. Some images even contain clear visual errors. I recommend that the authors conduct a more comprehensive assessment of image generation quality to better demonstrate the effectiveness of the proposed method.

2. The problem the paper aims to address is the low Mean Accepted Length that occurs when applying speculative decoding to AR image generation models. Specifically, although speculative decoding is used, a large number of predictions generated by the lightweight model are rejected, forcing the larger model to regenerate many tokens. As a result, the expected efficiency gains from speculative decoding are not realized. While the authors present a complete exploration from two perspectives, I still feel that, given the inherent ambiguity in token selection, improving the acceptance rate alone might suffice. A large set of tokens could be considered acceptable. This raises the question: Is speculative decoding still necessary in such a case? What would the quality and efficiency be if tokens were instead randomly generated within certain constraints?

**Questions:**

Please see the weaknesses.

---

> ### Author Response · Authors · 2024-11-21
> **Response to Reviewer 56i9 (1)**
>
> We deeply appreciate your positive evaluation of our work and your thoughtful feedback. Your recognition of the strengths in our methodology and the potential contributions of our approach is highly encouraging. Additionally, your detailed comments and constructive suggestions have been invaluable in guiding us to improve the quality and clarity of our work further. Below, we address each of the points you raised and provide additional experiments and explanations.
>
> **Weakness 1 : Evaluation of the image quality**
>
> We have conducted additional experiments using a broader set of evaluation metrics. The updated results are presented below (which is the same as in the general response), expanding on the main table in our original submission with same setting:
>
> | $\tau=1$ | Speedup | Mean Accepted Length | FID | CLIPScore | Precision / Recall | HPS v2 |
> | --- | --- | --- | --- | --- | --- | --- |
> | Vanilla AR | $\times1.00$ | $\times1.00$ | $15.22$ | $0.3203$ | $0.4781 / 0.5633$ | $24.11$ |
> | EAGLE-2 | $\times0.93$ | $\times1.20$ | - | - | - | - |
> | LANTERN ($\delta=0.10, k=1000)$ | $\times1.13$ | $\times1.75$ | $16.17$ | $0.3208$ | $0.4869 / 0.5172$ | $23.75$ |
> | LANTERN ($\delta=0.40, k=1000)$ | $\times1.69$ | $\times2.40$ | $18.76$ | $0.3206$ | $0.4909 / 0.4497$ | $23.22$ |
>
> As shown in the table above, LANTERN achieves comparable or slightly higher precision while exhibiting a modest reduction in recall. This suggests that the quality of individual images is preserved, albeit with a slight decrease in diversity.
>
> This modest decline in recall is likely due to the token selection ambiguity faced by the drafter. When the drafter is not optimally trained, it may struggle to generate predictions that are both diverse and accurate. As a result, increasing the acceptance probability to enhance acceleration could inadvertently reduce the diversity of the generated images. Nevertheless, this trade-off between speed and diversity has minimal impact on overall image quality, as demonstrated by stable FID, Precision and HPS v2 scores. Further improvements in drafter training could mitigate this effect, enhancing recall while maintaining the efficiency gains achieved by LANTERN. Additionally, the HPS v2 score, which is derived from a preference model trained on a human preference dataset, does not exhibit any significant degradation, further supporting the robustness of LANTERN's performance.
>
> We also understand your concern regarding the stylistic similarity and lack of text-image consistency in the generated samples. While we acknowledge this issue, we believe it primarily stems from the base visual AR model (LlamaGen Stage II) rather than being specific to LANTERN. To address this, we revised the text prompts used for generation of qualitative samples and have conducted additional experiments focusing on cases where the base model generates images that meet the following criteria: (1) stylistically dissimilar, (2) consistent with the text prompt, and (3) free of visual errors. For these cases, we provide samples generated by LANTERN to demonstrate its performance under such conditions. In addition, we provide samples generated by random replacement decoding as well. The qualitative samples are available at here : [LANTERN samples](https://postimg.cc/CBhYpPWQ), [random replacement decoding](https://postimg.cc/ygLK1hKY), and [qualitative samples](https://postimg.cc/6277FqK1). Each image have been incorporated into the Figure 1, 3, and 4 in the revised manuscript.
>
> We hope these additional results provide a more comprehensive perspective on the image quality generated by LANTERN.

---

> ### Author Response · Authors · 2024-11-21
> **Response to Reviewer 56i9 (2)**
>
> **Weakness 2 : Is speculative decoding still necessary in such a case?**
>
> We sincerely thank you for raising this important question. To explore the necessity of speculative decoding, we conducted an experiment using a randomly initialized drafter model to evaluate its ability to accelerate decoding. For this experiment, we utilized LlamaGen Stage I model as the target model and randomly sampled 1000 captions from the MS-COCO 2014 validation set to measure each model's performance. The results are summarized in the tables below:
>
> | $\tau=0$ | Mean Accepted Length |
> | --- | --- |
> | Random init drafter | $1.00$ |
> | trained drafter (4 epochs) | $1.54$ |
> | trained drafter (20 epochs) | $1.60$ |
>
> | $\tau=1$ | Mean Accepted Length |
> | --- | --- |
> | Random init drafter | $1.00$ |
> | trained drafter (4 epochs) | $1.18$ |
> | trained drafter (20 epochs) | $1.20$ |
>
> The results clearly demonstrate that a randomly initialized drafter model fails to improve the mean accepted length, achieving a value of 1.00 for both $\tau=0$ and $\tau=1$, equivalent to standard decoding without speculative decoding. In contrast, the trained drafter significantly enhances performance, achieving a mean accepted length of 1.60 and 1.20 for $\tau=0$ and $\tau=1$, respectively, after 20 epochs of training. These findings highlight the necessity of speculative decoding and the importance of having a well-trained drafter to achieve meaningful acceleration. Without proper training, the drafter cannot generate effective predictions, rendering speculative decoding ineffective. We are grateful for your insightful suggestion, as it allowed us to further validate the critical role of the drafter in speculative decoding and to provide a clearer understanding of its importance in our approach. This experiment reinforces the robustness and continued relevance of speculative decoding in achieving both speed and quality improvements.
>
> Once again, we thank you for your thoughtful comments and suggestions, which have significantly contributed to refining our work. We hope the additional experiments and explanations provided above adequately address your concerns. Please do not hesitate to reach out if further clarification or additional results are needed.

---

> ### Author Response · Authors · 2024-11-27
>
> Thank you for your kind words and for maintaining a positive recommendation for our research. We believe that your feedback has helped make our work even more robust.

---

### Official Review · Reviewer_v6x3 · 2024-11-02

**Soundness:** 2
**Presentation:** 2
**Contribution:** 2
**Rating:** 6
**Confidence:** 2

**Summary:**

In this paper, the authors propose LANTERN, a sampling strategy to speed up the image generation without losing too much quality. The method is implemented by accumulating probabilities from nearby tokens of the current sampling token. The authors further propose a thresholding technique to prevent the accumulated distribution deviating too much from the original one. Experiment results seem to be effective, however, the analysis is intuitive and empirical without deep insights.

**Strengths:**

- The proposed method is intuitive and easy to follow.

- Experimental results seem to verify the proposed method in terms of speeding up image generation without significant quality loss.

**Weaknesses:**

- There is a gap between the introduction and methodologies in writing. The related works are deferred to the appendix and problem definition and preliminaries such as Speculative Decoding are missing, resulting in difficulties in understanding the problem and challenge for readers who are not exactly working on this domain. On the other hand, Section 2.1 and 2.2 have some overlaps about the experiments and observation, which can be more concise.

- The evaluation seems to be insufficient. For the testing data, it is ideal to use the same setting as the vanilla baseline (i.e., LlamaGen) rather than just 100 captions from MSCOCO. The statement “Since measuring speedup with more than 100 samples shows no significant difference, we use 100 captions for efficiency” is not very convincing to me. In addition, other models are suggested for evaluation as well including other variations of LlamaGen.

- Some analyses in experiments are not sufficient. In sampling, the authors only mentioned that when $\delta$ is small, using larger k results in speeding up without significant degradation in performance, yet the reason for it is less explored. Section 4.3.2 also lacks further analysis why using TVD is better than JSD except some empirical results.

**Questions:**

- See the weakness.
- The claim of interchangeability in Section 3.1 is only empirically explored via a few examples. Are there some theoretical insights or statistics supporting this claim?

---

> ### Author Response · Authors · 2024-11-21
> **Response to Reviewer v6x3 (1)**
>
> We sincerely thank you for your thoughtful and constructive feedback. Your comments have highlighted critical areas for improvement and provided valuable insights that have strengthened our work. We are especially grateful for your positive remarks regarding the clarity and intuitiveness of our proposed method and its effectiveness in accelerating image generation. Below, we address each of the points raised with detailed responses.
>
> **Weakness 1: Gap between the introduction and methodologies in writing**
>
> We agree that the gap between the introduction and methodologies could have been improved. To address this, we revised the paper to include a concise preliminary section (Section 2 in the revised manuscript) in the main body, bridging the introduction and methodologies sections. Due to ICLR's page limit policy, we kindly ask for your understanding as we were unable to include the entire related work section. Additionally, Sections 2.1 and 2.2 have been merged into a single section (Section 3 in the revised manuscript) and refined to eliminate overlaps and improve conciseness. We believe these updates enhance the coherence of the paper, making it more accessible to readers and better aligning the introduction with the subsequent sections.
>
> **Weakness 2: Insufficiency in evaluating efficiency**
>
> First and foremost, we sincerely apologize for the misstatement in the original manuscript regarding the measurement of mean accepted length. While we had stated that the mean accepted length was measured on 100 images, this was incorrect. In fact, mean accepted lengths were evaluated on the full MS-COCO dataset, and this has been corrected in the revised manuscript.
>
> Additionally, we acknowledge that using only 100 captions to measure actual speedup (wall-clock time) may be insufficient for providing convincing evidence. To address this, we have re-measured the actual speedup using a larger sample size of 1000 captions in the revised version. We deeply regret any confusion caused by this oversight and thank the reviewers for bringing it to our attention, allowing us to improve the clarity and accuracy of the manuscript.
>
> | $\tau=0$ | Speedup | Mean Accepted Length |
> | --- | --- | --- |
> | Vanilla AR | $\times1.00$ | $\times1.00$ |
> | EAGLE-2 | $\times1.29$ | $\times1.60$ |
> | LANTERN ($\delta=0.05, k=1000)$ | $\times1.56$ | $\times2.02$ |
> | LANTERN ($\delta=0.20, k=1000)$ | $\times2.26$ | $\times2.89$ |
>
> | $\tau=1$ | Speedup | Mean Accepted Length |
> | --- | --- | --- |
> | Vanilla AR | $\times1.00$ | $\times1.00$ |
> | EAGLE-2 | $\times0.93$ | $\times1.20$ |
> | LANTERN ($\delta=0.10, k=1000)$ | $\times1.13$ | $\times1.75$ |
> | LANTERN ($\delta=0.40, k=1000)$ | $\times1.69$ | $\times2.40$ |
>
> In addition, we performed an analysis to confirm that speedup results stabilize as the number of captions increases. The table below illustrates this stability, showing that measurements with 1000 captions are consistent with those using larger sets:
>
> | Num Captions | Actual Speedup ($\tau=0$, LANTERN, $k=1000,0.05$) | Actual Speedup ($\tau=0$, LANTERN, $k=1000,0.2$) | Actual Speedup ($\tau=1$, LANTERN, $k=1000,0.1$) | Actual Speedup ($\tau=1$, LANTERN, $k=1000,0.4$) |
> | --- | --- | --- | --- | --- |
> | 100  | $1.56\times$ | $2.33\times$ | $1.13\times$ | $1.73\times$ |
> | 1000 | $1.56\times$ | $2.26\times$ | $1.13\times$ | $1.69\times$ |
> | 2000 | $1.57\times$ | $2.27\times$ | $1.13\times$ | $1.69\times$ |
> | 5000 | $1.56\times$ | $2.26\times$ | $1.13\times$ | $1.69\times$ |
>
>
> Please note that since the captions were randomly sampled, the results for the 100 captions may differ slightly from the acceleration reported in Table 3 of the original paper (Table 2 in the revised version). We updated our main result in Table 2 (in the revised version) with this re-measured speedup, and we hope these additional evaluations provide a clearer understanding of the robustness of our efficiency claims. The analysis of a number of captions has been added to Appendix F.1 in the revised manuscript.
>
> **Weakness 3: Further Analyses**
>
> - **Further analysis on $\delta$ and $k$**
>
>     As part of an extended evaluation prompted by another reviewer's question, we conducted additional experiments with image quality metrics (e.g., precision/recall and HPS v2) under the same settings as our main results. Through this process, we observed that increasing $k$ values does not consistently improve performance for small $\delta$, contrary to our initial claim. The table below illustrates this behavior:
>
>     | Configuration | Precision / Recall | HPS v2 |
>     | --- | --- | --- |
>     | $k=100, \delta=0.05$ | $0.4867$  / $0.5389$ | $24.01$ |
>     | $k=300, \delta=0.05$ | $0.4856$  / $0.5367$ | $23.97$ |
>     | $k=1000, \delta=0.05$ | $0.4865$  / $0.5334$ | $23.91$ |

---

> > ### Author Response · Authors · 2024-11-21
> > **Response to Reviewer v6x3 (2)**
> >
> > As shown in the table above, a decreasing trend in performance evaluation metrics can be observed as $k$ increases for the two new metrics. We sincerely apologize for the earlier claim that larger $k$ always improves speedup without significant quality degradation, as this does not hold consistently under these metrics. To address this, we have revised the manuscript to exclude this claim and provide a more accurate representation of the results. The updated findings have been appended to Appendix H in the revised manuscript. We thank the reviewers for highlighting this aspect, allowing us to refine our analysis and improve the clarity of the work.
> >
> > - **Further analysis on TVD and JSD**
> >
> >     To further address your suggestion, we compared TVD and JSD with respect to their impact on image quality and latency. The experiment setup is identical to our initial experiments on distance metrics. Actual computation time is evaluated in average on randomly selected 1000 samples. The results are as follows:
> >
> >     | Distance Metric | Mean Accepted Length | FID | CLIP Score |
> >     | --- | --- | --- | --- |
> >     | TVD ($\delta=0.3$) | $2.29\times$ | $18.27$ | $0.3206$ |
> >     | JSD ($\delta=0.2$) | $2.29\times$ | $18.21$ | $0.3206$ |
> >     | TVD ($\delta=0.2$) | $2.09\times$ | $17.43$ | $0.3208$ |
> >     | JSD ($\delta=0.13)$ | $2.09\times$ | $17.48$ | $0.3206$ |
> >
> >     | Distance Metric | Computation Time for Distance Metric | Total Computation Time of Single Decoding Step |
> >     | --- | --- | --- |
> >     | TVD | $1.19\times 10^{-3}$ s | $4.89\times 10^{-2}$s |
> >     | JSD | $4.03\times 10^{-3}$ s | $4.92 \times 10^{-2}$s |
> >
> >     In the table, it can be observed that when TVD and JSD yield similar mean accepted lengths, there’s no significant differences in terms of image quality. While either metric can be used without notable impact on image quality, TVD proves to be a more practical choice when considering the computation time. Since JSD requires more computation than TVD, selecting TVD is more beneficial for achieving speedup in practical applications. To validate this, we measured the computation time for TVD and JSD within a single decoding step of LANTERN. As shown in the table above, JSD requires more than three times the computation time of TVD. Although the time difference between these distance metrics is relatively small compared to the total time of whole decoding step, this difference accumulates over multiple decoding steps and can result in a significant impact on overall efficiency. The results discussed above have been incorporated into Table 3 and Table 9 in Appendix F.3 in the revised manuscript.
> >
> >
> > We hope that these additional analyses adequately address your concerns. Thank you for your insightful comments, which have guided us in refining our work.
> >
> > **Question 1 : Theoretical insights or statistical supports to the claim of interchangeability**
> >
> > Thank you for raising this critical question. We acknowledge that empirical qualitative evidence based on a few examples is insufficient to fully support the claim of interchangeability. To address this, we have conducted additional experiments to provide statistical evidence. Specifically, we evaluated the quality of generated images using random replacement decoding and compared them to standard decoding. We used LlamaGen Stage I model and evaluate FID and CLIP Score on MS-COCO 2017 validation set.
> >
> > | Randomly Replaced by one of $k$-th nearest token | FID | CLIP Score |
> > | --- | --- | --- |
> > | Vanilla AR | $25.06$ | $0.3214$ |
> > | $k=50$ | $26.88$ | $0.3120$ |
> > | $k=100$ | $30.76$ | $0.3091$ |
> > | $k=1000$ | $88.03$ | $0.2715$ |
> >
> > The results show that as $k$ increases, the replaced token is selected from a broader set of latent space neighbors, but the image quality remains well-preserved up to a certain threshold. For $k=50$, the FID increases slightly from $25.06$ (Vanilla AR) to $26.88$, and the CLIP Score decreases marginally from $0.3214$ to $0.3120$, indicating minimal degradation. Similarly, for $k=100$, the FID rises moderately to $30.76$, and the CLIP Score drops slightly to $0.3091$, demonstrating that even with $k=100$, the image quality remains stable and acceptable.
> >
> > It is only at $k=1000$ that a significant decline becomes apparent, with the FID increasing sharply to $88.03$ and the CLIP Score dropping to $0.2715$, highlighting the negative impact of selecting tokens from more distant neighbors. These results confirm our earlier qualitative observations in Figure 3 in the revised manuscript that increasing $k$ up to 100 maintains reasonable image quality, making it a viable strategy for generative tasks. This highlights the robustness of the model in preserving image fidelity under controlled token replacement within this range.

---

> > > ### Author Response · Authors · 2024-11-21
> > > **Response to Reviewer v6x3 (3)**
> > >
> > > The result has been included in the revised manuscript, at Appendix C.1. We believe these findings strengthen the validity of our claim and provide a more robust foundation for this observation. Additionally, we aim to position the development of theoretical insights into interchangeability as a promising direction for future research. Your feedback has been invaluable in highlighting the need for a deeper exploration of this aspect, and we are sincerely grateful for your thoughtful comments.
> > >
> > > Once again, we would like to express our heartfelt gratitude for your detailed review and constructive suggestions. Your feedback has not only helped us identify areas for improvement but has also motivated us to refine our work further. We hope that the additional experiments, analyses, and revisions we have outlined above adequately address your concerns. Please do not hesitate to let us know if there are any other aspects we should clarify or improve. We sincerely value your time and effort in reviewing our submission and are deeply appreciative of the insights you have provided.

---

### Author Response · Authors · 2024-11-21
**General Response**

We sincerely thank the reviewers for their thoughtful feedback and valuable suggestions. Your insights have been instrumental in improving the clarity and quality of our work, and we greatly appreciate the opportunity to address your concerns. In this response, we have included a **General Response** for questions raised by two or more reviewers, ensuring consistency and transparency. For individual responses, we have provided tailored responses, with relevant excerpts from the general response included where appropriate.

---
### **Broader range of image quality evaluation**

To further validate LANTERN's ability to maintain image quality, we conducted additional evaluations using two text-to-image quality metrics: (1) Precision and Recall [1], and (2) Human Preference Score (HPS) v2 [2], with same setting as our main experiment. The results below extend our original main table (Table 3 in original paper, Table 2 in the revised version) to include these metrics. We are currently working on the evaluation for greedy decoding ($\tau=0$), and the results will be updated as soon as available.

| $\tau=1$ | Speedup | Mean Accepted Length | FID | CLIPScore | Precision / Recall | HPS v2 |
| --- | --- | --- | --- | --- | --- | --- |
| Vanilla AR | $\times1.00$ | $\times1.00$ | $15.22$ | $0.3203$ | $0.4781$ / $0.5633$ | $24.11$ |
| EAGLE-2 | $\times0.93$ | $\times1.20$ | - | - | - | - |
| LANTERN ($\delta=0.10, k=1000)$ | $\times1.13$ | $\times1.75$ | $16.17$ | $0.3208$ | $0.4869$  / $0.5172$ | $23.75$ |
| LANTERN ($\delta=0.40, k=1000)$ | $\times1.69$ | $\times2.40$ | $18.76$ | $0.3206$ | $0.4909$  / $0.4497$ | $23.22$ |

As shown in the table above, LANTERN demonstrates comparable or slightly improved precision while showing a slight decrease in recall. This can be interpreted as maintaining the quality of individual images while slightly reducing diversity. This slight reduction in recall can be attributed to the challenges associated with token selection ambiguity in the drafter. When the drafter is suboptimally trained, it may struggle to provide sufficiently diverse predictions or precise alternatives. Consequently, the increased acceptance probability, while improving acceleration, can lead to a modest decline in the diversity of generated images. However, it is important to note that this trade-off between acceleration and diversity does not significantly impact the overall image quality, as evidenced by the consistent FID, Precision and HPS v2 scores. Future work could address this by improving the drafter’s training process to better handle token selection ambiguity, thereby further enhancing recall without compromising LANTERN’s efficiency. Additionally, the HPS v2 score, which is derived from a preference model trained on a human preference dataset, does not exhibit any significant degradation, further supporting the robustness of LANTERN's performance. As a result, LANTERN prove its effectiveness over various metric, by maintaining its score in acceptable range. The results have been updated to the Table 2 in the revised version.

---
### **Evaluation on other visual AR models**

To evaluate LANTERN on additional visual AR models, we extended our experiments to include the LlamaGen Stage II model and Anole [3]. These experiments focused on random sampling ($\tau=1$) for the following reasons: (1) Random sampling generally produces higher-quality images than greedy decoding ($\tau=0$), (2) Our prior results (Table 3 in the paper) indicate random sampling is a more challenging case for acceleration, and (3) Resource constraints prevented comprehensive experiments for both settings. We use MS-COCO 2017 Validation set to evaluate LlamaGen stage II and Anole’s image generation performance. In addition, we measure actual speedup of LlamaGen model with RTX3090 and A100 for Anole. The results are summarized below.

| LlamaGen Stage II, $\tau=1$ | Speedup | Mean Accepted Length | FID | CLIPScore | Precision / Recall | HPS v2 |
| --- | --- | --- | --- | --- | --- | --- |
| Vanilla AR | $1.00\times$ | $1.00\times$ | $47.60$ | $0.2939$ | $0.4138$  / $0.5648$ | $23.84$ |
| EAGLE-2 | $0.96\times$ | $1.22\times$ | - | - | - | - |
| LANTERN ($\delta=0.40, k=1000)$ | $1.64\times$ | $2.24\times$ | $46.10$ | $0.2925$ | $0.4704$  / $0.5222$ | $23.06$ |

| Anole, $\tau=1$ | Speedup | Mean Accepted Length | FID | CLIPScore | Precision / Recall | HPS v2 |
| --- | --- | --- | --- | --- | --- | --- |
| Vanilla AR | $1.00\times$ | $1.00\times$ | $20.27$ | $0.3215$ | $0.6552$  / $0.6398$ | $23.52$ |
| EAGLE-2 | $0.73\times$ | $1.10\times$ | - | - | - | - |
| LANTERN ($\delta=0.50, k=100)$ | $1.17\times$ | $1.83\times$ | $23.40$ | $0.3186$ | $0.6026$  / $0.6178$ | $22.92$ |

---

> ### Author Response · Authors · 2024-11-21
> **General Response (Continued)**
>
> Anole exhibits significantly lower next-token prediction probabilities (average top-$1$ probability of $0.064$ and average top-$10$ probability of $0.204$) compared to LlamaGen ($0.206$ and $0.520$, respectively), as shown in Figure 2 (c) of the revised manuscript. These results indicate that Anole suffers from more severe token selection ambiguity, which inherently affects the drafter’s training process. Consequently, Anole’s drafter achieves a test accuracy of only $27.60\%$, markedly lower than the $38.80\%$ test accuracy observed for LlamaGen’s drafter, as shown in Figure 2 (b) in the revised manuscript. This discrepancy in drafter performance directly impacts the effectiveness of both EAGLE-2 and LANTERN on Anole.
>
> It is important to emphasize that this variation does not reflect a limitation of LANTERN itself but rather underscores the influence of model-specific characteristics, such as token selection ambiguity, on overall performance. Despite these challenges, LANTERN continues to outperform EAGLE-2 on Anole in terms of acceleration while maintaining acceptable image quality, highlighting its robustness across diverse models.
>
> Furthermore, these findings suggest a promising direction for future research. Addressing token selection ambiguity through improved drafter architectures or incorporating ambiguity-aware training techniques could enhance the drafter’s performance, further bolstering LANTERN’s effectiveness across models with severe token selection ambiguity. This indicates that while the current results demonstrate LANTERN's robustness and general applicability, there is potential for even greater improvements with advancements in drafter training and model design.
>
> We have included the these results to appendix E in the revised manuscript, as we aimed to minimize significant changes to the main content of the manuscript. However, if the reviewers collectively feel that incorporating these results into the main table would enhance clarity and comprehensiveness, we would be happy to expand the main table in the next revision. We sincerely appreciate all of your valuable feedback and remain committed to improving the manuscript in line with your suggestions.
>
>
> ---
> ### **Major changes in the revised manuscript**
>
> Again, we sincerely thank the reviewers for their valuable suggestions and thoughtful feedback, which have greatly contributed to improving the quality of our work. Based on these insights, we have carefully revised the manuscript to address the concerns raised and incorporate additional analyses.
>
> We have made several key revisions to enhance the clarity, organization, and comprehensiveness of the manuscript:
>
> 1. **Preliminaries Section (Section 2)**: Added a new section to bridge the Introduction and Methodologies, providing essential background on visual AR models and speculative decoding to improve reader understanding.
> 2. **Consolidation of Sections 2.1 and 2.2**: These sections were merged and rewritten to eliminate overlapping content and improve the flow of information. While the text was significantly revised to achieve a clearer and more concise presentation, the underlying content and key messages remain unchanged.
> 3. **Expanded Experimental Section**: Incorporated additional results using new evaluation metrics into the main table (Table 2) and corresponding explanations in the Experimental Setup (Section 5.1) and Results sections.
>     - Updates in ablation study : To provide a clear comparison between TVD and JSD, we compared the differences in image quality between TVD and JSD when both achieved the same level of acceleration. This analysis has been reflected in Table 3 and Section 5.3.2 of the revised manuscript.
> 4. **Updated Qualitative Samples**: Replaced the samples in Figures 1, 3, and 4 with more representative examples to better align with text prompts, address visual errors, and showcase stylistic diversity. The updated figures also can be found in these anonymous links: [LANTERN samples](https://postimg.cc/CBhYpPWQ), [random replacement decoding](https://postimg.cc/ygLK1hKY), and [qualitative samples](https://postimg.cc/6277FqK1).
>
> We hope these additional experiments further support the contributions of our work and provide clarity on its robustness. Once again, thank you for your time and effort in reviewing our submission. We are grateful for your constructive feedback and look forward to your continued insights.
>
> **References**
>
> [1] Kynkäänniemi, Tuomas, et al. "Improved precision and recall metric for assessing generative models." *Advances in neural information processing systems* 32 (2019).
>
> [2] Wu, Xiaoshi, et al. "Human preference score v2: A solid benchmark for evaluating human preferences of text-to-image synthesis." *arXiv preprint arXiv:2306.09341* (2023).
>
> [3] Chern, Ethan, et al. "Anole: An open, autoregressive, native large multimodal models for interleaved image-text generation." *arXiv preprint arXiv:2407.06135* (2024).

---

> ### Author Response · Authors · 2024-11-27
> **General Response (Updated)**
>
> As previously mentioned, we further evaluate the performance of LANTERN under greedy decoding ($\tau=0$) and summarize the results in the following table. Similar to the case of sampling ($\tau=1$), we report both acceleration and image quality across various metrics. Similar to the sampling, the findings demonstrate that LANTERN achieves substantial speedup while maintaining competitive image quality compared to standard AR decoding in the greedy decoding as well. These results reaffirm LANTERN's capability to balance efficiency and quality across different decoding strategies. This additional analysis has also been incorporated into the main table (Table 2) of the revised manuscript.
>
> | $\tau=0$ | Speedup | Mean Accepted Length | FID | CLIPScore | Precision / Recall | HPS v2 |
> | --- | --- | --- | --- | --- | --- | --- |
> | Vanilla AR | $\times1.00$ | $\times1.00$ | $28.63$ | $0.3169$ | $0.4232$ / $0.3517$ | $23.18$ |
> | EAGLE-2 | $\times1.29$ | $\times1.60$ | - | - | - | - |
> | LANTERN ($\delta=0.05, k=1000)$ | $\times1.56$ | $\times2.02$ | $29.77$ | $0.3164$ | $0.4484$ / $0.3158$ | $22.62$ |
> | LANTERN ($\delta=0.20, k=1000)$ | $\times2.26$ | $\times2.89$ | $30.78$ | $0.3154$ | $0.4771$ / $0.2773$ | $21.69$ |

---

### Author Response · Authors · 2024-12-03
**Gratitude and Final Remarks on Paper #12935**

Dear Reviewers, AC, and SAC of paper #12935,

As we approach the conclusion of the author-reviewer discussion phase, we would like to highlight few points:

1. We are sincerely grateful to the reviewers for their constructive feedback, which has significantly contributed to improving the clarity and depth of the manuscript. The thoughtful comments and suggestions have been invaluable in refining our work.

2. During the discussion period, we made every effort to address all concerns raised by the reviewers, including conducting additional experiments, collecting data, and providing thorough responses. While we are glad that this process helped garner more favorable assessments from some reviewers, we regret that two reviewers were unable to engage further despite our polite reminders. Out of respect for their time, we chose not to send repeated reminders, trusting that our detailed rebuttal and the positive feedback from other reviewers sufficiently highlight the merits of our work. Nevertheless, if circumstances allow, we would still greatly appreciate their feedback, which could further enrich the evaluation process.

3. Understanding the limitations and potential for speculative decoding (SD) for vision AR models has been an open problem. Our paper "LANTERN" not only takes the first attempt to understand the limitations of SD for vision AR but also identifies the associated potential causes and presents a novel solution to mitigate that. We demonstrate LANTERN can yield inference speed up by up to $1.82\times$ compared to any existing alternative. We believe this work would inspire the community to adopt SD for the emerging vision AR use cases for latency-critical applications.

We remain hopeful and trustful in the reviewing system and grateful for the reviews and discussions (though only two out of four reviewers provided them). We hope our efforts are acknowledged and taken into account during the decision.

Best regards,

The Authors

Paper ID #12935

---

### Meta-Review · Area_Chair_weKp · 2024-12-24

**Metareview:**

The paper introduces LANTERN, a method that accelerates visual autoregressive models by adapting speculative decoding -- a mechanism originally proposed for large language models -- to the domain of visual autoregressive generation. The proposed approach demonstrates the ability to achieve speed gains while maintaining image quality. This work opens up new possibilities in enhancing the efficiency of autoregressive visual generative models and highlights how mechanisms from large language models can be effectively tailored for visual-specific applications. Considering the positive feedback from all reviewers, I recommend the acceptance of this paper.

**Additional Comments On Reviewer Discussion:**

The authors effectively addressed the concerns raised by the reviewers by incorporating additional metrics (e.g., Precision/Recall and HPS v2) to comprehensively evaluate image quality, conducting analyses to validate theoretical claims such as token interchangeability and the advantages of TVD over JSD, and making clarifications and improvements to the paper’s structure and coherence.

---

### Decision · Program_Chairs · 2025-01-22

Accept (Poster)